# Data-Adaptive Exposure Thresholds under Network Interference

**Vydhourie Thiyageswaran**
Department of Statistics
University of Washington
Seattle, WA, USA
vrtt@uw.edu

**Tyler H. McCormick**
Department of Statistics
University of Washington
Seattle, WA, USA
tylermc@uw.edu

**Jennifer Brennan**
Google Research
Kirkland, WA, USA
jrbrennan@google.com

## Abstract

Randomized controlled trials often suffer from interference, a violation of the Stable Unit Treatment Value Assumption (SUTVA), where a unit's outcome is influenced by its neighbors' treatment assignments. This interference biases naive estimators of the average treatment effect (ATE). A popular method to achieve unbiasedness pairs the Horvitz-Thompson estimator of the ATE with a known exposure mapping, a function that identifies units in a given randomization unaffected by interference. For example, an exposure mapping may stipulate that a unit experiences no further interference if at least an $h$-fraction of its neighbors share its treatment status. However, selecting this threshold $h$ is challenging, requiring domain expertise; in its absence, fixed thresholds such as $h = 1$ are often used. In this work, we propose a data-adaptive method to select the $h$-fractional threshold that minimizes the mean-squared-error (MSE) of the Horvitz-Thompson estimator. Our approach estimates the bias and variance of the Horvitz-Thompson estimator paired with candidate thresholds by leveraging a first-order approximation, specifically, linear regression of potential outcomes on exposures. We present simulations illustrating that our method improves upon non-adaptive threshold choices, and an adapted Lepski's method. We further illustrate the performance of our estimator by running experiments with synthetic outcomes on a real village network dataset, and on a publicly-available Amazon product similarity graph. Furthermore, we demonstrate that our method remains robust to deviations from the linear potential outcomes model.

## 1 Introduction

Estimating the Average Treatment Effect (ATE), the difference in the average outcomes of units when all units are treated versus when none are, is challenging in the presence of network interference. Under interference, the Stable Unit Treatment Value Assumption (SUTVA) [Cox, 1958, Rubin, 1978, Manski, 1990] is violated, as a unit's outcome is influenced by its neighbors' treatment assignments. This problem is salient in randomized controlled trials, the setting we examine. For example, estimating the adoption of a new idea through random assignment of people to advertisements is complicated by the dissemination of information through social interactions. Consider a social media platform seeking to estimate the efficacy of a product innovation in increasing user engagement. Measuring outcomes for users in the control group is complicated by their interactions with treated

39th Conference on Neural Information Processing Systems (NeurIPS 2025).

users. If the innovation successfully increases engagement among treated users, their friends in the control group may also exhibit increased engagement through their interactions on the platform. As such discrete interactions can be modeled through network interference models, the problem of estimating average treatment effects under network interference becomes a ubiquitous one.

A fundamental challenge arises from the lack of knowledge about the exact interference pattern. *Exposure mappings*, as defined by Aronow and Samii [2017], are functions that partition the space of treatment assignments and individual-level features (e.g., social neighborhood structure) into distinct exposure values. These mappings encapsulate the concept of "effective treatments" introduced in [Manski, 2013]. For example, let $z \in \{0,1\}^n$ denote the treatment assignment vector, and $W \in [0,1]^{n \times n}$ represent the (weighted) adjacency matrix, where $W_{ij}$ encodes the relationship between units $i$ and $j$. A simple exposure mapping takes the form $f \colon (z, W) \mapsto Wz$, assigning each unit a weighted sum of its neighbors' treatments. Aronow and Samii [2017], Ugander et al. [2013], Sussman and Airoldi [2017], Hardy et al. [2019], Auerbach and Tabord-Meehan [2021] expound upon different exposure mappings, and estimation under these settings. Following [Eckles et al., 2017, Toulis and Kao, 2013], we characterize exposure in terms of the fraction of treated neighbors. For instance, a unit may be considered as experiencing no additional interference if at least an $h$-fraction of its neighbors share its treatment assignment. This fractional neighborhood exposure mapping aligns with the fractional thresholds model [Watts, 1999], where treated and untreated peers exert opposing influences on a unit's outcome. Centola and Macy [2007] illustrate this with the example of refraining from littering in a neighborhood: an individual's decision to avoid littering depends on the relative number of neighbors who also refrain from littering.

Under a known $h$-fractional neighborhood exposure mapping, the Horvitz-Thompson estimator [Horvitz and Thompson, 1952] is a popular unbiased estimator of the average treatment effect. Without domain knowledge of the interference structure, however, such estimators based on non-adaptive treatment exposure conditions often suffer from high bias or variance. For instance, the Horvitz-Thompson estimator can be paired with an extreme threshold $h$ to achieve unbiasedness, requiring that a unit and all its neighbors be treated (or in control). However, this can result in high variance, since the probability of such configurations is very low, especially for high-degree nodes. Conversely, setting a low threshold $h$ introduces bias into the estimator by including units with limited treatment (resp. control) exposure in the treatment (resp. control) exposed group. We propose a simple data-dependent approach to find an MSE-optimal threshold for the Horvitz-Thompson estimator in the finite population setting. We compute the *bias slope*, an approximate rate of change of bias, which we use to estimate the bias of our estimator. We use this information together with variance estimates to select a threshold minimizing the Mean-Squared-Error (MSE) of the estimator.

In [Basse and Airoldi, 2018, Cai et al., 2015], a (known) generalized linear form of network effects on neighborhood treatments was assumed. Related to our work, Zhu et al. [2024] fit functionals on the treatment assignment and exposure. In [Chin, 2019], the author proposes regression adjustment estimators that predict potential outcomes under global treatment and control conditions. Unlike traditional regression adjustments, their approach constructs adjustment variables from functions of the treatment assignment vector, framing the learning of a more flexible exposure mapping as a feature engineering problem. While Belloni et al. [2022] focus on estimating direct effects under network interference, their approach to determining the exposure radius, i.e. the $m_i$-hop influence, shares conceptual ground with our approach of selecting optimal exposure thresholds.

In this paper, we present our framework for the Horvitz-Thompson estimator of the ATE, motivated by its widespread adoption in practice. Nevertheless, our approach could be adapted for other estimators, like the Hájek, Difference-in-Means, and Augmented Inverse Propensity Weighted (AIPW) estimators. We formulate the adaptively-thresholded estimator for the Difference-in-Means estimator, which is a special case of the Hájek estimator, in the Appendix A.7.5. We note however, that the Hájek estimator is only approximately unbiased at the true threshold. This makes formal characterization of its exact bias–variance tradeoff more challenging than for the Horvitz-Thompson estimator.

Within off-policy evaluation in reinforcement learning, Su et al. [2020] address the classical problem of adaptive bandwidth selection, well studied in non-parametric statistics [Fan and Gijbels, 1992, Ruppert, 1997, Kallus and Zhou, 2018, Mukherjee et al., 2015], by applying Lepski's method [Lepskii, 1992, 1993, Lepski and Spokoiny, 1997, Goldenshluger and Lepski, 2011]. Our work explores an extension of this framework to optimal bandwidth selection for the Horvitz-Thompson estimator in average treatment effect estimation under network interference (see Appendix A.3). To

our knowledge, this connection has not been explored previously. We compare our procedure to an adaptation of Lepski's method tailored to our problem setting. In our context, the network dependence structure can lead to violations of the monotonicity and decay rates conditions that Lepski's method relies on. Our approach offers a simple and principled data-adaptive alternative designed to address these challenges in our context. Philosophically, these ideas are similar. This perspective on threshold, or equivalently bandwidth, selection at the estimation stage can loosely be viewed as the "dual" of choosing cluster sizes in cluster-randomization approaches such as Ugander et al. [2013], Eckles et al. [2017] at the design stage.

## 2 Setup

### 2.1 Notation.

We use $h$ to denote the threshold for "treatment exposure", and $1-h$ for "control exposure". Therefore, for larger $h$, i.e. $h$ closer to one, we have a more restrictive setting, where only the subset of treated (resp. control) units with most of their neighbors treated (resp. control) are counted as treatment (resp. control) exposed. For smaller $h$, i.e. $h$ close to zero, we have a less restrictive setting, as a larger subset of treated (resp. control) units satisfy the threshold condition. Throughout the paper we use $W$ to denote the adjacency matrix with $W_{ii} = 0$, $d_i = \sum_j W_{ij}$ for the degree of node $i$, and $d$ when the graph is regular. We use $D$ to denote the diagonal matrix of degrees $d_i$.

### 2.2 Problem Setup

We study the finite population setting. Additionally, we consider a fractional neighborhood exposure mapping. That is, for any treatment assignment vector $z \in \{0,1\}^n$, the effective influence of the graph's treatment assignment on the outcome of node $i$ is equivalent to the influence of fractional treatment assignments in the neighborhood of node $i$. This reduces our potential outcomes model to the following form:

$$y_i(z) = y_i(z_i, e_i) = \alpha_i + \psi(z_i, e_i) + \epsilon_i, \tag{1}$$

where $z_i \in \{0,1\}$ is the treatment assignment of unit $i$, $e_i \in [0,1]$ is the fraction of treated neighbors of unit $i$ (excluding $i$ itself), and the $\epsilon_i$ are uniformly-bounded and non-random. Throughout the paper, we focus on $\psi(z_i, e_i) = g(z_i) + f(e_i)$, but our framework generalizes. For instance, we will consider the linear potential-outcome model in some examples:

$$y_i(z) = \alpha_i + \psi(z_i, e_i) + \epsilon_i = \alpha_i + \beta_i z_i + \gamma_i e_i + \epsilon_i. \tag{2}$$

We are interested in estimating the average treatment effect (ATE),

$$\tau = \frac{1}{n} \sum_{i=1}^n y_i(1,1) - \frac{1}{n} \sum_{i=1}^n y_i(0,0) = \frac{1}{n} \sum_{i=1}^n (\alpha_i + \beta_i + \gamma_i + \epsilon_i - \alpha_i - \epsilon_i) = \frac{1}{n} \sum_{i=1}^n (\beta_i + \gamma_i), \tag{3}$$

though we could, more generally, take any difference between exposure categories.

Let $Y_i$ denote the observed outcome for unit $i$, $i \in [n]$. We consider the Horvitz-Thompson estimator [Horvitz and Thompson, 1952] for a given exposure threshold $h$:

$$\hat{\tau}_h = \frac{1}{n} \sum_{i=1}^n \frac{\mathbf{1}\{z_i = 1, e_i \geq h\}}{\mathbb{P}\{z_i = 1, e_i \geq h\}} Y_i - \frac{1}{n} \sum_{i=1}^n \frac{\mathbf{1}\{z_i = 0, e_i \leq 1-h\}}{\mathbb{P}\{z_i = 0, e_i \leq 1-h\}} Y_i \tag{4}$$

Our goal then is to select the threshold $h$ in the Horvitz-Thompson estimator above that minimizes the MSE. More formally, given a candidate set of thresholds $H$, we select

$$h^* := \underset{h \in H}{\operatorname{argmin}} \operatorname{MSE}(\hat{\tau}_h) = \underset{h \in H}{\operatorname{argmin}} \operatorname{Bias}^2(\hat{\tau}_h) + \operatorname{Var}(\hat{\tau}_h). \tag{5}$$

Stronger interference, i.e. when $\psi(\cdot, e_i)$ is more sensitive to changes in exposure $e_i$, induces greater bias through edges that connect units with differing treatment assignments. To mitigate this bias, one

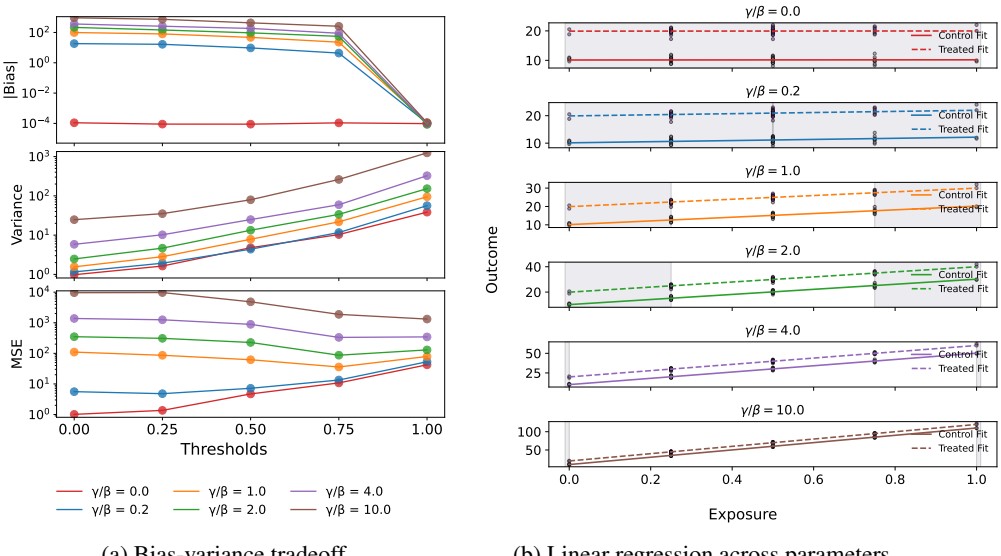

| (a) Bias-variance tradeoff | (b) Linear regression across parameters |

Figure 1: Left: bias, variance, and RMSE of AdaThresh across different thresholds for the 1000-node 2nd-power cycle graph (see Appendix A.6) with unit-level Bernoulli randomization, and linear model outcome $Y_i = 10 + 10z_i + \gamma e_i + \epsilon_i$, for fixed $\epsilon_i$ generated from $\mathcal{N}(0,1)$. Right: toy illustration of linear fits used to approximate exposure bias caused by including more data across different thresholds. The "jittered" datapoints signify the amount of variance reduction across thresholds. Rectangular blocks depict the data regions under the MSE-optimal thresholds.

might pick a higher exposure threshold $h$. However, increasing $h$ reduces the exposure probabilities $\mathbb{P}\{z_i = 1, e_i \geq h\}$ and $\mathbb{P}\{z_i = 0, e_i \leq 1 - h\}$, thereby increasing variance. To illustrate this bias–variance trade-off, we display the biases and variances under the linear model $Y_i = \alpha + \beta z_i + \gamma e_i + \epsilon_i$ evaluated across varying thresholds in the Horvitz-Thompson estimator (4) above, in the left panel of Figure 1. This trade-off is illustrated for various ratios of $\gamma/\beta$. The right panel of Figure 1 illustrates the different fitted linear models for various $\gamma/\beta$ ratios, whose slopes we call the bias slope. We then use these bias slopes to compute the bias estimates for $\{\hat{\tau}_h\}_h$ (4), as described in detail in the next section. Combining these bias estimates with the corresponding variance estimates allows us to select the MSE-optimal threshold, shown by the shaded regions in the right panel of Figure 1 (matching the left panel).

We adopt the MSE as our objective as we are interested in precisely and accurately estimating the ATE, trading off bias against variance in the estimator. See Deng et al. [2024, Section 2.2] for more discussions on this. We note that, while the overall trend of bias decreasing and variance increasing with $h$ holds, neither quantity is necessarily monotonic due to the dependence between units and, consequently, their exposures. To further illustrate this, in Appendix A.6, we draw upon toy examples of circulant graphs from Ugander et al. [2013].

## 3 Estimating bias and variance

With the goal of obtaining an MSE-optimal threshold in mind, we find the *bias slope* $\hat{\gamma}_n$, which captures how the bias changes with the threshold $h$. More concretely, the bias slope quantifies the average change in bias as $h$ is varied from 0 to 1, estimated by the slope of a linear fit. Generally, this linear regression coefficient $\hat{\gamma}_n$ is simply a first-order approximation of how $\psi(\cdot, e_i)$ changes with $e_i$, estimating the average change in the mean outcome per unit change in exposure. In our context, this also reflects the average change in bias as we move away from the boundary exposures, i.e., from 0 for control or 1 for treated, since in our setting bias arises when exposures deviate from these extremes. This interpretation motivates the use of the weights $e_i - 0 = e_i$ and $1 - e_i$ in Equation 6, corresponding to deviations from the control and treated boundaries, respectively.

Building on this interpretation, we propose the following Horvitz-Thompson-style estimator for the bias, which integrates the bias slope over the empirical exposure distribution to yield an average (absolute) bias estimate at a given threshold:

$$\hat{b}(\hat{\tau}_h) = \frac{1}{n} \sum_{i=1}^{n} \frac{(1 - e_i)\hat{\gamma}_n \mathbf{1}\{z_i = 1, e_i \geq h\}}{\mathbb{P}\{z_i = 1, e_i \geq h\}} + \frac{1}{n} \sum_{i=1}^{n} \frac{e_i \hat{\gamma}_n \mathbf{1}\{z_i = 0, e_i \leq 1 - h\}}{\mathbb{P}\{z_i = 0, e_i \leq 1 - h\}}, \quad (6)$$

where $\hat{\gamma}_n$, the bias slope, is the linear regression coefficient of the outcome on the exposure variable. If the potential outcomes depend strongly and positively on the treatment exposures, $\hat{\gamma}_n$ will be larger. Consequently, our estimator will capture a more pronounced increase in bias caused by including more data when using a smaller threshold $h$. We use a Horvitz-Thompson-style bias estimator to account for the exposure distribution, which may pose challenges for Lepski's method (see description in Appendix 12).

To estimate the variance associated with a threshold $h$, we use the variance estimator discussed in Aronow and Samii [2017], available in Appendix A.1.

Together, they inform the choice of threshold, and our adaptive estimator, AdaThresh. We describe our procedure in Algorithm 1.

---

**Algorithm 1** AdaThresh

---

**Require:** Graph adjacency matrix $W$, outcome vector $Y$, treatment vector $z$
1: Compute exposure: $e \leftarrow D^{-1}Wz$
2: Fit linear model: $Y = \beta z + \gamma e + c$
3: Let $\hat{\gamma}$ be the estimated coefficient for $e$
4: **for** each threshold $h \in \mathcal{H}$ **do**
5:     Estimate bias: $\widehat{\text{Bias}}(h)$ using $\hat{\gamma}, Y, z, W$, and $h$            ▷ See Eq. (6)
6:     Estimate variance: $\widehat{\text{Var}}(h)$ using $Y, z, W$, and $h$            ▷ See Appendix A.1
7:     Compute $\widehat{\text{MSE}}(h) \leftarrow \widehat{\text{Bias}}^2(h) + \widehat{\text{Var}}(h)$
8: **end for**
9: $\hat{h} \leftarrow \arg\min_{h \in \mathcal{H}} \widehat{\text{MSE}}(h)$
10: **return** $\hat{\tau}_{\hat{h}}$            ▷ See (4)

---

The bias slope from the linear fit captures the average change in bias as $h$ is varied from 0 to 1. We do not assume an underlying linear model $Y = \alpha + \beta z + \gamma e$, for which the sum of linear regression coefficients $\hat{\beta} + \hat{\gamma}$ would be the optimal estimate of the ATE. Rather, we use a linear regression only to obtain the bias slope, a first-order approximation of how $\psi(\cdot, e_i)$ changes with $e_i$. This idea is similar in spirit to the work of Baird et al. [2018], where the authors leverage the slope of spillovers with respect to treatment saturation levels as an approximation, without assuming linearity in the true data-generating process.

As we first estimate the bias slope to inform our overall estimator of the ATE, sample-splitting is one potential strategy for consistent estimation. However, since the units of interest are dependent (as modeled by the network interference), one would have to prove performance guarantees by leveraging results such as Hart and Vieu [1990]. In our setting, however, it is not necessary that we split the data as we leverage the simplicity of the model class [Van Der Vaart et al., 1996]. In Appendix A.9, we discuss this using Donsker results [Van Der Vaart et al., 1996] to demonstrate that our approach does not lead to overfitting.

## 4 Theoretical Results

Denote by $H$ the (finite) set of exposure thresholds. For example, in cycle graphs (see Appendix A.6 for details), $H = \{\frac{u}{d} : u \in \{0, 1, ..., d\}\}$ where $d$ is the degree of the graph. We assume the following:

**Assumption 4.1** (Unconfoundedness, Positivity)**.**

1. (Unconfoundedness) For all $i \in [n]$,

$$\{y_i(\tilde{z}_i, \tilde{e}_i) : \tilde{z}_i \in \{0, 1\}, \tilde{e}_i \in [0, 1]\} \perp\!\!\!\perp (z_i, e_i)$$

2. (Positivity) For all $i \in [n]$, $h \in [0, 1]$,

$$\mathbb{P}\{z_i = 1, \, e_i \geq h\} > 0 \quad \text{and} \quad \mathbb{P}\{z_i = 0, \, e_i \leq 1 - h\} > 0.$$

The first part of Assumption 4.1 states that the individual and neighborhood treatment assignments are independent of the potential outcomes. This follows from the unit-level and cluster-level Bernoulli randomization designs we consider in this paper. The second part of the assumption is also a fundamental one in the causal inference literature [Rosenbaum and Rubin, 1983, Petersen et al., 2012], ensuring that we have sufficient data to estimate the ATE. It also relates to statistical leverage [Mahoney et al., 2011, Martinsson and Tropp, 2020, Young, 2019, Pilanci and Wainwright, 2015]. The idea, informally, is that under positivity, statistical leverage across the input data is more homogeneous. As a result of this, there are no observations that have too much control over the linear regression.

**Assumption 4.2** (Bounded variation treatment-exposure function)**.** Let the potential outcome function be $y_i = \alpha_i + \psi(z_i, e_i) + \epsilon_i$ as in (1). The function $\psi(z_i, e_i)$ has bounded variation.

**Assumption 4.3** (Bounded First-Order Interactions)**.** For unit-level randomization, we assume that all nodes have bounded degree. Specifically, there exists a finite constant $d_{\max} < \infty$ such that, for every unit $i \in [n]$, the degree $d_i$ satisfies

$$d_i \leq d_{\max}.$$

For cluster-level randomization, we assume that all clusters have bounded degrees of cross-cluster interactions. Let $s_i$ denote the number of cross-cluster connections involving the cluster to which unit $i$ belongs. Then, there exists a finite constant $s_{\max} < \infty$ such that, for every cluster $i \in \mathcal{C}$, the cross-cluster degree $s_i$ satisfies

$$s_i \leq s_{\max}.$$

These restrictions are consistent with the observation that networks in practice tend to be sparse and clustered [Barabási, 2013, Chandrasekhar, 2016, Chandrasekhar et al., 2020]. If this assumption is violated, exposure positivity may fail, and Proposition A.5 will no longer hold. However, as discussed in Remark 4.4 below, this assumption can be weakened under certain conditions. Some examples of graph classes that fall under the categorization of Assumption 4.3 are expander graphs, and growth-restricted graphs [Alon, 1986, Arora et al., 2009, Gkantsidis et al., 2003, Kuhn et al., 2005, Krauthgamer and Lee, 2003, Kowalski, 2019]. This is related to requiring the graph interference structure to have small $k_n$-conductance (i.e. the $k_n$-partition generalization of the Cheeger constant) for $k_n$ growing with $n$. In the causal inference literature, the growth-restricted graph setting was studied in [Ugander et al., 2013].

*Remark* 4.4. Assumption 4.3 is also employed by Aronow and Samii [2017]. As we further discuss in Appendix A.8, it can be relaxed by "binning" the exposures and verifying whether the "affinity set" conditions of Chandrasekhar et al. [2023] are satisfied. Specifically, in more general weighted settings, one can partition exposures into bins indexed by $b = 1, 2, \ldots, K$, each associated with an effective exposure level $\tilde{e}_b$. In this framework, node degrees need not be bounded, provided the affinity set conditions are met. These conditions subsume the approximate neighborhood interference (ANI) condition of Leung [2022] when the maximum clique size in the network remains bounded. For cases with growing clique size, we refer the reader to Leung [2022] for further discussion of the ANI framework.

In the following theorem, we characterize the probability of choosing the correct threshold under the best average linear fit, for which the exposure slope is $\gamma^*$. The correct threshold $h^*$ under the best average linear fit is the threshold minimizing the sum of the true variance and the true squared-bias under $Y_i = \alpha + \beta z_i + \gamma^* e_i$. We write $M_n^*(h), \hat{M}_n(h)$ to denote the MSE under the true average best fit line and the MSE estimated by our methods, respectively, at threshold $h$ and finite-population size $n$.

**Theorem 4.5.** *Suppose Assumptions 4.1, 4.2, and 4.3 hold. Further assume unit-level Bernoulli randomization with treatment assignment probability p. Let $\Delta_n := \min_{h \in H} |M_n^*(h) - M_n^*(h^*)|$, and let $U_n \in \mathbb{R}$ satisfy $\max_h |b_n^*(\hat{\tau}_h)| \leq U_n$, which exists by Assumptions 4.1(2) and 4.2. Denote by $h_n^*$ the optimal threshold under the true average best-fit line, and by $\hat{h}_n$ the threshold chosen by our*

*method. Finally, let $H$ be the set of exposures. Then,*

$$\mathbb{P}\{\hat{h}_n \neq h_n^*\} \leq \sum_{h \in H} \mathbb{P}\{|\hat{M}_n(h) - M_n^*(h)| > \Delta_n/2\}$$

$$\leq 3|H| \max \left\{ \exp\left( -\frac{\Delta_n np(1-p)}{8cd_{\max}} + 1 \right), \quad \exp\left( -\frac{\Delta_n^2 np(1-p)}{(16U_n)^2 cd_{\max}} + 1 \right), \right.$$

$$\left. 6 \exp\left( -\frac{Cn\left(\frac{\Delta_n}{4} - \frac{cd_{\max}^2}{n}\right)}{\sqrt{A_{n,2}} + \sqrt{\left(\frac{\Delta_n}{4} - \frac{cd_{\max}^2}{n}\right)M_{n,2}}} \right) \right\},$$

*for some constants $c, C$. Here, $A_{n,p} \leq 16\|\tilde{v}\|_{L_1}^2 \{c_1 + \frac{(\log n)^4}{n}\}^2$, for some constant $c_1$ and $\tilde{v}$ is the Fourier transform of the summands of the variance components, and $M_{n,2} = 4\|\tilde{v}\|_{L_1}(\log n)^4$.*

*If, instead, the design is cluster-level Bernoulli randomization with probability $p$, denote by $s_{\max}$ the maximum cross-cluster degree, i.e., the largest number of distinct clusters to which any given cluster is connected. Then, we can replace $d_{\max}$ by $s_{\max}$ in the bounds above.*

In Theorem 4.5, we assume without loss of generality that the MSE gap is lower-bounded by a constant, i.e. $\Delta_n \geq \Delta$, as otherwise all candidate thresholds are optimal. The proof to the theorem is in Appendix A.5.1. In this proof, we make use of the results from [Ziemann et al., 2024, Theorem 3.1], and [Shen et al., 2020, Theorem 2.1].

Our results tell us that generally, for fixed $n$, and under a unit-level Bernoulli randomization, our approach yields a higher probability of choosing the optimal threshold when the maximum degree is smaller. Under a Bernoulli cluster-level randomization, our approach yields a higher probability of choosing the optimal threshold when the maximum cross-cluster degree is smaller. Therefore, for a well-clusterable graph, our approach would yield a higher probability of choosing the optimal threshold under cluster-level randomization, as the clustering would give us a reduction in the cross-cluster degree. Not surprisingly, when $n$ is larger, and the degree ranges are kept fixed, the probability of choosing the optimal threshold is higher. Additionally, as the variance of the variance estimator is larger, the probability of picking the right threshold is smaller. The error scales with the size of the bias, appearing in the second term in the bound. Finally, considering symmetric $h$ for treated and control units, our approach yields a higher probability of choosing the optimal threshold when the treatment assignment probability $p = 0.5$. This can be seen from the $p(1-p)$ term in the bound which is maximized at $p = 0.5$.

This gives us the following corollary. Denote the true MSE by $M_n^{**}$, and the corresponding bias and optimal threshold by $b_n^{**}$ and $h_n^{**}$, respectively.

**Corollary 4.6.** *Under the conditions of Theorem 4.5, further assume that $\psi(z_i, e_i) = g(z_i) + f(e_i)$, and that $\sup_{e_i} |f(e_i) - \gamma^* e_i| \leq \delta$. Let $U_n^* \in \mathbb{R}$ be such that $\max_h |b_n^{**}(\hat{\tau}_h)| \leq U_n^*$, which exists by Assumptions 4.1 (2) and 4.2. Note that $U_n^*$ is bounded from below. Let $\tilde{\delta} := 16\delta^2 + 8\delta U_n^*$. Then,*

$$\mathbb{P}\{\hat{h}_n \neq \hat{h}_n^{**}\} \leq \sum_{h \in H} \mathbb{P}\{|\hat{M}_n(h) - M_n^{**}(h)| > \Delta_n/2\}$$

$$\leq 3|H| \max \left\{ \exp\left( -\frac{(\Delta_n/8 - \tilde{\delta})np(1-p)}{cd_{\max}} + 1 \right), \right.$$

$$\exp\left( -\frac{(\Delta_n^2/(16U_n)^2 - \tilde{\delta})np(1-p)}{cd_{\max}} + 1 \right),$$

$$\left. 6 \exp\left( -\frac{Cn\left(\frac{\Delta_n}{4} - \frac{cd_{\max}^2}{n} - \tilde{\delta}\right)}{\sqrt{A_{n,2}} + \sqrt{\left(\frac{\Delta_n}{4} - \frac{cd_{\max}^2}{n} - \tilde{\delta}\right)M_{n,2}}} \right) \right\},$$

*for some constants $c, C$. Here, $A_{n,p} \leq 16\|\tilde{v}\|_{L_1}^2 \{c_1 + \frac{(\log n)^4}{n}\}^2$, for some constant $c_1$ and $\tilde{v}$ is the Fourier transform of the summands of the variance components, and $M_{n,2} = 4\|\tilde{v}\|_{L_1}(\log n)^4$.*

*If, instead, the design is cluster-level Bernoulli randomization with probability $p$, denote by $s_{\max}$ the maximum cross-cluster degree, i.e., the largest number of distinct clusters to which any given cluster is connected. Then, we can replace $d_{\max}$ by $s_{\max}$ in the bounds above.*

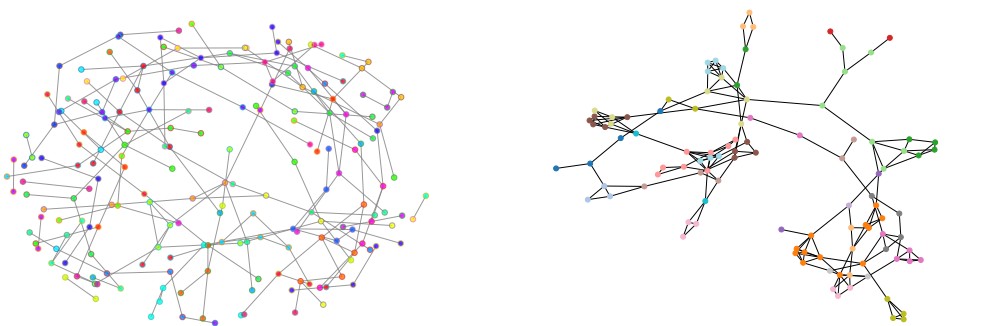

Figure 2: Left: A graph of size $n = 200$ generated from the Stochastic Block Model (SBM). Node colors reflect ground-truth cluster membership, corresponding to the underlying block membership. Right: Village network dataset, with node colors reflecting $\epsilon = 1$-net clustering (see [Ugander et al., 2013] for details on $\epsilon$-net clustering).

Corollary 4.6 tells us that if the maximum linear approximation error between the best average linear fit and the true exposure function $f(\cdot)$ is small enough relative to the minimum MSE gap $\Delta_n$, our estimator will be optimal with high probability for large $n$. Indeed, when the $n(\Delta_n - \tilde{\delta})$ terms are large, the $\exp\left(-n(\Delta_n - \tilde{\delta})\right)$ terms are small. If $f(x) = \gamma x$ for all $x \in [0, 1]$ indeed, then with necessarily we have that with high probability for large $n$, our estimator is optimal.

## 5 Experiments

We now compare the performance of our approach with that of non-adaptive Horvitz-Thompson estimators $HT(1)$ and $HT(0)$, with $h = 1$, and $h = 0$, respectively. Additionally, since our objective is essentially an MSE-optimal bandwidth selection problem for the indicator kernel (see Appendices A.3, 13), we also compare our estimator to a variant where the threshold is selected via Lepski's method [Goldenshluger and Lepski, 2011, Su et al., 2020]. We note that Lepski's method relies on the monotonicity of the bias and the variance to work well. In our setting, due to the implicit dependency structure from the graph appearing in the exposures $e_i$, the monotonicity assumption may be violated. We write out these three estimators we compare against in Appendix A.2.

In Figure 3, we display simulation results with outcomes generated from the linear model with $\psi(z_i, e_i) = g(z_i) + f(e_i)$, $\alpha_i = 10$, $g(z_i) = \beta z_i = 10z_i$, $f(e_i) = \gamma e_i$, with fixed $\epsilon_i$ generated from $\mathcal{N}(0, 1)$, for a 200-node graph, generated from an SBM with 40 underlying clusters (see Figure 2 (left panel)). We focus on varying the ratio $\gamma/\beta$ as we consider a fixed graph. We see that our adaptive thresholding estimator, AdaThresh, generally performs better than existing estimators, interpolating between the fixed 0/1-threshold Horvitz-Thompson estimators, and out-performing the Lepski-based Horvitz-Thompson estimators when the threshold $\gamma/\beta$ is high. We also display simulation results for 1000-node 2nd-power cycle graphs in Figure 7.

Furthermore, cluster-randomized designs better control interference, reducing bias over wider ranges of spillover ratios. In such settings, the variance tends to dominate the bias. Since optimizing for bias leads to $HT(1)$, and optimizing for variance leads to $HT(0)$, we observe better intermediate thresholds for longer ranges of spillover ratios under cluster-randomized designs compared to unit-level designs. These patterns become even more pronounced in the cycle graphs presented in Figure 7, where the node degree is held constant. From the design perspective (see, for instance, [Viviano et al., 2023]), if spillover ratios are larger, then one might analogously perform more cluster-randomized designs to contain more of the interference and reduce bias.

### 5.1 Real Data

We evaluate the performance of our estimator on village (No.6) network data from [Banerjee et al., 2013]. Here, $n = 110$, with nodes representing villagers and edges indicating that the adjacent

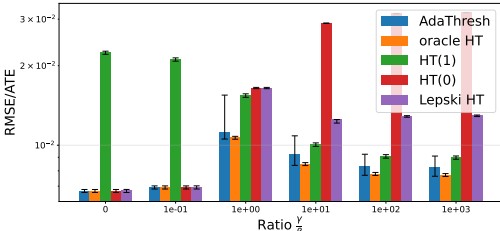
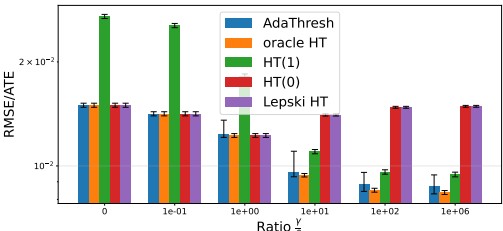

Figure 3: RMSE (normalized by the ATE) of different Horvitz-Thompson estimators for the SBM graph in Figure 2 (left panel). Left: unit-level Ber(0.5) randomization. Right: cluster-level Ber(0.5) randomization with ground truth clusters. The error bars are the empirical 95% confidence intervals around the mean.

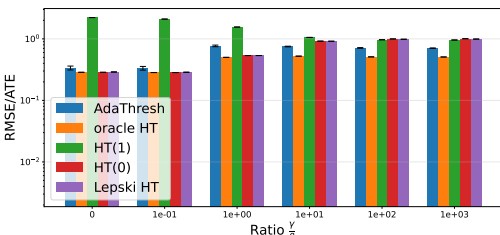
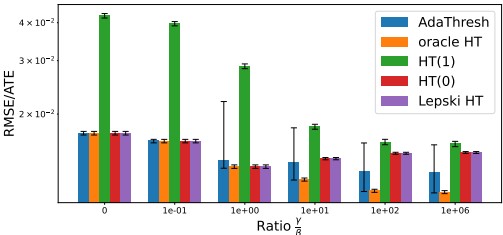

Figure 4: RMSE (normalized by the ATE) of different Horvitz-Thompson estimators for the village network in Figure 2 (right panel). Left: unit-level Ber(0.5) randomization. Right: cluster-level Ber(0.5) randomization with $\epsilon(=1)$-net clustering. The error bars are the empirical 95% confidence intervals around the mean.

villagers have visited each other's homes. We ran experiments with synthetic potential outcomes, averaging over 1000 trials, under unit-level and cluster-level Bernoulli randomizations, where clusters were formed using an $\epsilon(=1)$-net clustering (see [Ugander et al., 2013] for details on $\epsilon$-net clustering). We generate simulated data using the linear model with $\psi(z_i, e_i) = g(z_i) + f(e_i)$, $\alpha_i = 10, g(z_i) = \beta z_i = 10 z_i, f(e_i) = \gamma e_i$, with fixed $\epsilon_i$ generated from $\mathcal{N}(0,1)$. To compute the exposure probabilities, we used $2 \times 10^4$ Monte-Carlo trials. We focus on varying the ratio $\gamma/\beta$ as we consider a fixed graph. Figure 4 demonstrates that our method generally interpolates between HT(1) and HT(0), achieving intermediate performance on smaller graphs (i.e., smaller $n$). While it does not consistently outperform the HT(0) baseline across all settings, the results suggest that our approach remains competitive across a large range of regimes and offers a reasonable bias–variance trade-off. We hypothesize that the observed imprecision and inaccuracy of our estimator stems from inaccuracies in estimating smaller exposure probabilities, particularly for higher-degree nodes. For larger $n$ and smaller $d_{\max}$, performance improves further, supporting our theoretical findings, as demonstrated in Appendix A.7 on the Amazon (DVD) products similarity network [Leskovec et al., 2007] (see Figure 6), and on various circulant graphs.

## 6 Discussion

In this paper, we focused on the additive model for $\psi$, but note that our framework is applicable more broadly. We investigated AdaThresh, an adaptive Horvitz-Thompson estimator for symmetric thresholds $h$ and $1-h$ for treatment and control, respectively. Our framework applies more generally with estimator thresholds $h_1$ and $1 - h_0$, respectively with $h_1 \neq h_0$. Additionally, in Appendix A.7.5, we demonstrate the performance of this approach using the Difference-in-Means estimator incorporating exposure thresholds. We leave it to future work to extend our results to more general exposure estimation procedures. In [Chin, 2019], the author proposes learning the feature variables that are most predictive of the potential outcomes. One could extend our work by first learning the feature variables, as proposed by [Chin, 2019], followed by then learning the appropriate optimal threshold associated with these features, using our approach.

In Appendix A.7.4, we illustrate the robustness of our estimator to deviations from linearity in the potential outcomes model. We note that our framework would apply in broader settings, including a direct extension to friends-of-friends (or further connections) or to neighborhoods that are either observed or learned through a clustering algorithm. An additional interesting future direction could be to extend our work to the framework presented in Chandrasekhar et al. [2023], allowing for a complete graph underlying the interference structure.

Furthermore, to improve upon robustness to non-linear settings, we propose an extension using local regression to estimate the rate of change of bias within the $[0, 1-h], [h, 1]$ windows. As long as there is sufficient concentration of exposures in these windows, robustness is never worse. In Appendix A.7.6, we display our simulation results in this setting using local linear regression to estimate the rate of change of bias within the $[0, 1-h], [h, 1]$ windows. One could also use other local regression approaches, such as kernel regression, etc., while maintaining sufficiently small model complexity to avoid needing to sample-split.

This paper prioritizes minimizing the mean squared error (MSE) to effectively balance the bias–variance trade-off in a Horvitz-Thompson-type estimator, rather than focusing on inference. However, we acknowledge the significance of characterizing inferential tools, such as confidence intervals, in specific contexts. Notably, [Aronow and Samii, 2017] and [Athey et al., 2018] offer valuable insights into inferential methods for estimating average treatment effects under network interference.

## Acknowledgements

The authors would like to thank Alex Kokot and Lars van der Laan for helpful discussions; Dean Eckles for pointing us to a reference, and Matthew Eichhorn for suggesting the Amazon product similarity network dataset.

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

# A    Appendix / supplemental material

## Notation

In this supplement, we write $b^*(\tau_h), \hat{\tau}_b(h), v^*(\tau_h), \hat{v}(\tau_h)$ to represent the true bias, estimated bias, true variance, and estimated variance, respectively, at threshold $h$. We also write $d_{\max}$ to represent the upper-bound (Assumption 4.3) on the degrees of the nodes in the network.

## A.1    Variance estimator

We use the variance estimator, for the Horvitz-Thompson estimator, proposed by [Aronow and Samii, 2017, Eqn. 7,9].

$$
\begin{aligned}
\hat{v}(\hat{\tau}_h) = & \sum_{i=1}^{n} \frac{\mathbf{1}\{z_i = 1, e_i \geq h\}Y_i^2}{n^2 \pi_i^1}\left(\frac{1}{\pi_i^1} - 1\right) \\
& + \sum_{i=1}^{n}\sum_{\substack{j=1 \\ j \neq i}}^{n} \frac{\mathbf{1}\{z_i = 1, e_i \geq h\}\mathbf{1}\{z_j = 1, e_j \geq h\}Y_iY_j}{n^2 \pi_{ij}^{11}}\left(\frac{\pi_{ij}^{11}}{\pi_i^1 \pi_j^1} - 1\right) \\
& + \sum_{i=1}^{n} \frac{\mathbf{1}\{z_i = 0, e_i \leq 1 - h\}Y_i^2}{n^2 \pi_i^0}\left(\frac{1}{\pi_i^0} - 1\right) \\
& + \sum_{i=1}^{n}\sum_{\substack{j=1 \\ j \neq i}}^{n} \frac{\mathbf{1}\{z_i = 0, e_i \leq 1 - h\}\mathbf{1}\{z_j = 0, e_j \leq 1 - h\}Y_iY_j}{n^2 \pi_{ij}^{00}}\left(\frac{\pi_{ij}^{00}}{\pi_i^0 \pi_j^0} - 1\right) \\
& - \frac{2}{n^2}\sum_{i=1}^{n}\sum_{j\in[n];\pi_{ij}^{10}>0} \left(\mathbf{1}\{z_i = 1, e_i \geq h\}\mathbf{1}\{z_j = 0, e_j \leq 1 - h\}Y_iY_j\right)\left(\frac{1}{\pi_i^1 \pi_j^0} - \frac{1}{\pi_{ij}^{10}}\right) \\
& + \frac{2}{n^2}\sum_{i=1}^{n}\sum_{j\in[n];\pi_{ij}^{10}=0} \left(\frac{\mathbf{1}\{z_i = 1, e_i \geq h\}Y_i^2}{2\pi_i^1} + \frac{\mathbf{1}\{z_j = 0, e_j \leq 1 - h\}Y_j^2}{2\pi_j^0}\right)
\end{aligned}
\tag{7}
$$

Here, $\pi_i^x, \pi_{ij}^{xy}$ are defined with respect to the threshold $h$. That is, $\pi_i^1 = \mathbb{P}\{z_i = 1, e_i \geq h\}$, $\pi_i^0 = \mathbb{P}\{z_i = 0, e_i \leq 1 - h\}$, $\pi_{ij}^{10} = \mathbb{P}\{z_i = 1, e_i \geq h, z_j = 0, e_j \leq 1 - h\}$, $\pi_{ij}^{11} = \mathbb{P}\{z_i = 1, e_i \geq h, z_j = 1, e_j \geq h\}$, $\pi_{ij}^{01} = \mathbb{P}\{z_i = 0, e_i \leq 1 - h, z_j = 1, e_j \geq h\}$, $\pi_{ij}^{00} = \mathbb{P}\{z_i = 0, e_i \leq 1 - h, z_j = 0, e_j \leq 1 - h\}$.

## A.2    Horvitz-Thompson estimators with existing approaches

We write the vanilla Horvitz-Thompson estimator (at threshold one) 8, the Horvitz-Thompson estimator at threshold zero 9, and the Horvitz-Thompson estimator with the threshold selected via Lepski's method 12 below.

### A.2.1    Vanilla Horvitz-Thompson estimator (at threshold one)

$$
\hat{\tau}_{\mathrm{HT}_1} = \frac{1}{n}\sum_{i=1}^{n}\frac{\mathbf{1}\{z_i = 1, e_i = 1\}}{\mathbb{P}\{z_i = 1, e_i = 1\}}Y_i - \frac{1}{n}\sum_{i=1}^{n}\frac{\mathbf{1}\{z_i = 0, e_i = 0\}}{\mathbb{P}\{z_i = 0, e_i = 0\}}Y_i
\tag{8}
$$

### A.2.2    Horvitz-Thompson estimator at threshold zero

$$
\hat{\tau}_{\mathrm{HT}_0} = \frac{1}{n}\sum_{i=1}^{n}\frac{\mathbf{1}\{z_i = 1\}}{\mathbb{P}\{z_i = 1\}}Y_i - \frac{1}{n}\sum_{i=1}^{n}\frac{\mathbf{1}\{z_i = 0\}}{\mathbb{P}\{z_i = 0\}}Y_i
\tag{9}
$$

### A.2.3    Lepski's method

As described in [Lepski and Spokoiny, 1997], we first take

$$
I(h) := [\hat{\tau}_{\mathrm{HT}_h} - 2\widehat{\mathrm{SDEV}}(\hat{\tau}_{\mathrm{HT}_h}), \hat{\tau}_{\mathrm{HT}_h} + 2\widehat{\mathrm{SDEV}}(\hat{\tau}_{\mathrm{HT}_h})].
\tag{10}
$$

Then, take

$$
\hat{h}_{\mathrm{Lepski}} := \min\{h \in H : \cap_{h' \in H : h' \geq h}I(h') \neq \emptyset\},
\tag{11}
$$

and

$$\hat{\tau}_{\text{LepskiHT}} = \frac{1}{n} \sum_{i=1}^{n} \frac{\mathbf{1}\{z_i = 1, e_i \geq \hat{h}_{\text{Lepski}}\}}{\mathbb{P}\{z_i = 1, e_i \geq \hat{h}_{\text{Lepski}}\}} Y_i - \frac{1}{n} \sum_{i=1}^{n} \frac{\mathbf{1}\{z_i = 0, e_i \leq 1 - \hat{h}_{\text{Lepski}}\}}{\mathbb{P}\{z_i = 0, e_i \leq 1 - \hat{h}_{\text{Lepski}}\}} Y_i \tag{12}$$

Lepski's method requires monotonicity and decay rate conditions to be satisfied. Our setting may lead to violations of these conditions.

## A.3 An equivalent formulation of the Horvitz-Thompson estimator with the exposure threshold

We can rewrite the display in 4 in the following form:

$$\hat{\tau}_h = \frac{1}{n} \sum_{i=1}^{n} \frac{\mathbf{1}\{|z_i - e_i| \leq 1 - h\}}{\mathbb{P}\{|z_i - e_i| \leq 1 - h \mid z_i\}} \tilde{z}_i Y_i, \tag{13}$$

where $\tilde{z}_i = 2z_i - 1$, so that $\tilde{z}_i \in \{-1, 1\}$. It is then clear that the threshold $h$ controls how much dissimilarity between the treatment status of units and their neighbors, we allow in our estimation. This allows us to reframe the problem as an optimal bandwidth selection one.

## A.4 Bias and Variance estimation errors

In this subsection, we write down the estimation errors of the bias and the variance terms in the MSE estimation.

For the bias terms, we have,

$$\hat{b}(\hat{\tau}_h) - b^*(\hat{\tau}_h) = \left( \frac{1}{n} \sum_{i=1}^{n} \frac{\hat{\gamma}_n (1 - e_i) \mathbf{1}\{z_i = 1, e_i \geq h\}}{\mathbb{P}(z_i = 1, e_i \geq h)} - \frac{1}{n} \sum_{i=1}^{n} \sum_{x_i \in X_i} \frac{\mathbf{1}\{x_i \geq h\}}{|x_i : x_i \in X_i \cap x_i \geq h|} \gamma_n^*(1 - x_i) \right)$$

$$+ \left( \frac{1}{n} \sum_{i=1}^{n} \frac{\hat{\gamma}_n e_i \mathbf{1}\{z_i = 0, e_i \leq 1 - h\}}{\mathbb{P}(z_i = 0, e_i \leq 1 - h)} - \frac{1}{n} \sum_{i=1}^{n} \sum_{x_i \in X_i} \frac{\mathbf{1}\{x_i \leq 1 - h\}}{|x_i : x_i \in X_i \cap x_i \leq 1 - h|} \gamma_n^*(x_i) \right),$$

where $\gamma$ is the slope of the best average linear fit, and where $x_i$ ranges over the set $X_i$ of possible fractions of degree $i$.

By an abuse of notation, we use $y_i(h^+)$ to represent the average (across possible exposure fractions for node $i$) potential outcome for unit $i$ with exposures that are at least $h$, while we use $y_i(h^-)$ to represent the average (across possible exposure fractions for node $i$) potential outcome for unit $i$ with exposures that are at most $1 - h$.

In the variance terms,

$$v^*(\hat{\tau}_h) = \sum_{i=1}^{n} \frac{y_i(h^+)^2}{n^2} \left( \frac{1}{\pi_i^1} - 1 \right)$$

$$+ \sum_{i=1}^{n} \sum_{\substack{j=1 \\ j \neq i}}^{n} \frac{y_i(h^+) y_j(h^+)}{n^2} \left( \frac{\pi_{ij}^{11}}{\pi_i^1 \pi_j^1} - 1 \right)$$

$$+ \sum_{i=1}^{n} \frac{y_i(h^-)^2}{n^2 \pi_i^0} \left( \frac{1}{\pi_i^0} - 1 \right)$$

$$+ \sum_{i=1}^{n} \sum_{\substack{j=1 \\ j \neq i}}^{n} \frac{y_i(h^-) y_j(h^-)}{n^2} \left( \frac{\pi_{ij}^{00}}{\pi_i^0 \pi_j^0} - 1 \right)$$

$$- \frac{2}{n^2} \sum_{i=1}^{n} \sum_{j \in [n]; \pi_{ij}^{10} > 0} y_i(h^+) y_j(h^-) \left( \frac{\pi_{ij}^{10}}{\pi_i^1 \pi_j^0} - 1 \right)$$

$$+ \frac{2}{n^2} \sum_{i=1}^{n} \sum_{j \in [n]; \pi_{ij}^{10} = 0} y_i(h^+) y_j(h^-).$$

Therefore, from the above and Section A.1, we have that

$$\sup_h \hat{v}(\hat{\tau}_h) - v^*(\hat{\tau}_h)$$

$$= \sup_h \left[ \left( \sum_{i=1}^n \frac{\mathbf{1}\{z_i = 1, e_i \ge h\}Y_i^2}{n^2 \pi_i^1} \left( \frac{1}{\pi_i^1} - 1 \right) - \sum_{i=1}^n \frac{y_i(h^+)^2}{n^2} \left( \frac{1}{\pi_i^1} - 1 \right) \right) \right.$$

$$+ \left( \sum_{i=1}^n \sum_{\substack{j=1 \\ j \ne i}}^n \frac{\mathbf{1}\{z_i = 1, e_i \ge h\}\mathbf{1}\{z_j = 1, e_j \ge h\}Y_i Y_j}{n^2 \pi_{ij}^{11}} \left( \frac{\pi_{ij}^{11}}{\pi_i^1 \pi_j^1} - 1 \right) \right.$$

$$\left. - \sum_{i=1}^n \sum_{\substack{j=1 \\ j \ne i}}^n \frac{y_i(h^+)y_j(h^+)}{n^2} \left( \frac{\pi_{ij}^{11}}{\pi_i^1 \pi_j^1} - 1 \right) \right)$$

$$+ \left( \sum_{i=1}^n \frac{\mathbf{1}\{z_i = 0, e_i \le 1 - h\}Y_i^2}{n^2 \pi_i^0} \left( \frac{1}{\pi_i^0} - 1 \right) - \sum_{i=1}^n \frac{y_i(h^-)^2}{n^2 \pi_i^0} \left( \frac{1}{\pi_i^0} - 1 \right) \right)$$

$$+ \left( \sum_{i=1}^n \sum_{\substack{j=1 \\ j \ne i}}^n \frac{\mathbf{1}\{z_i = 0, e_i \le 1 - h\}\mathbf{1}\{z_j = 0, e_j \le 1 - h\}Y_i Y_j}{n^2 \pi_{ij}^{00}} \left( \frac{\pi_{ij}^{00}}{\pi_i^0 \pi_j^0} - 1 \right) \right.$$

$$\left. - \sum_{i=1}^n \sum_{\substack{j=1 \\ j \ne i}}^n \frac{y_i(h^-)y_j(h^-)}{n^2} \left( \frac{\pi_{ij}^{00}}{\pi_i^0 \pi_j^0} - 1 \right) \right)$$

$$- \frac{2}{n^2} \sum_{i=1}^n \sum_{\substack{j \in [n] \\ \pi_{ij}^{10} > 0}} \left( \mathbf{1}\{z_i = 1, e_i \ge h\}\mathbf{1}\{z_j = 0, e_j \le 1 - h\}Y_i Y_j \left( \frac{1}{\pi_i^1 \pi_j^0} - \frac{1}{\pi_{ij}^{10}} \right) \right.$$

$$\left. - y_i(h^+)y_j(h^-) \left( \frac{\pi_{ij}^{10}}{\pi_i^1 \pi_j^0} - 1 \right) \right)$$

$$+ \frac{2}{n^2} \sum_{i=1}^n \sum_{\substack{j \in [n] \\ \pi_{ij}^{10} = 0}} \left( \frac{\mathbf{1}\{z_i = 1, e_i \ge h\}Y_i^2}{2\pi_i^1} + \frac{\mathbf{1}\{z_j = 0, e_j \le 1 - h\}Y_j^2}{2\pi_j^0} \right.$$

$$\left. \left. - y_i(h^+)y_j(h^-) \right) \right].$$

## A.5 Proofs to Theorem 4.5 and Corollary 4.6

### A.5.1 Proof to Theorem 4.5

*Proof to Theorem 4.5.* We first consider the variance terms. Define $\bar{v} = \mathbb{E}[\hat{v}]$. We have that,

$$\mathbb{P}(|\hat{v} - v^*| > \Delta_n/4) = \mathbb{P}(|\hat{v} - \bar{v} + \bar{v} - v^*| > \Delta_n/4)$$

$$\stackrel{(a)}{=} \mathbb{P}(|\hat{v} - \bar{v}| > \Delta_n/4 - cd_{\max}^2/n)$$

$$\stackrel{(b)}{\le} 6 \exp \left( - \frac{Cn \left( \frac{\Delta_n}{4} - \frac{cd_{\max}^2}{n} \right)}{\sqrt{A_{n,2}} + \sqrt{\left( \frac{\Delta_n}{4} - \frac{cd_{\max}^2}{n} \right) M_{n,2}}} \right)$$

where $A_{n,p} \le 16\|\tilde{v}\|_{L_1}^2 \{c_1 + \frac{(\log n)^4}{n}\}^2$, for some constant $c_1$ and $\tilde{v}$ is the Fourier transform of the variance summands, and $M_{n,2} = 4\|\tilde{v}\|_{L_1}(\log n)^4$.

The equality $(a)$ is obtained as a result of Assumption 4.3. Indeed, we know that $\bar{v} - v^*$ is non-zero only when there exists $i, j$ such that $\pi_{ij}(h^+, h^-) = 0$, i.e. the joint exposure assignment probability of $e_i \ge h$, and $e_i \le 1 - h$. Since the contribution of these to $\bar{v} - v^*$ are bounded above by $cd_{\max}^2/n$ for some constant $c$, we get $(a)$. The inequality $(b)$ is obtained from a direct application of [Shen et al., 2020, Theorem 2.1], as it easy to see that our exposure dependencies satisfy $\alpha$-mixing conditions and that together with Assumption 2, the variance summands satisfy the Fourier transform conditions of [Shen et al., 2020, Theorem 2.1].

Next, we consider the bias terms, for any constant $J$,

$$\mathbb{P}((\hat{\gamma}_n - \gamma_n^*)^2 > \Delta_n/J) \leq \exp\left(-\frac{\Delta_n np(1-p)}{Jcd_{\max} + 1}\right),$$

for some constant $c$. We use the results from [Ziemann et al., 2024, Theorem 3.1]. Below, we verify that the conditions are met in our setting. We begin with the condition that for every $v \in \mathbb{R}$ such that $v = 1/(\mathbb{E}[e_i^2])^{1/2}$, there exists $h \in \mathbb{R}^+$ such that, $\mathbb{E}[(ve_i)^4] \leq h^2 v^2 \mathbb{E}[e_i^2]$. This is trivially satisfied in our setting since our exposures is bounded. Next, we consider the first part of [Ziemann et al., 2024, Condition (3.3)]. This is also trivially satisfied in our setting since our block sizes ($= d_{\max}^2$) are bounded by Assumption 4.3. The second part of [Ziemann et al., 2024, Condition (3.3)] is also satisfied since our de-meaned noise-class interaction variables (as defined in [Ziemann et al., 2024, Eq.(2.5)]) are sub-gaussian by assumption of independence between the potential outcomes' residuals and the treatment design. Next, [Ziemann et al., 2024, Condition (3.4)] is also trivially satisfied since we take our blocks to be of size $d_{\max}^2$ almost, with the possible exception of the final remaining block, uniformly. Finally, it is easy to see that [Ziemann et al., 2024, Condition (3.5)] is also satisfied since the exposure variables are independent outside of the radius $d_{\max}^2$, since exposure variables $e_i$ and $e_j$ for any $i, j \in [n]$ are only dependent when nodes $i$ and $j$ share a neighbor. We quickly note that the monotone partitioning (as described in [Ziemann et al., 2024, Theorem 3.1] is not necessary in our setting, as the "blocking" procedure introduced by [Yu, 1994] holds more generally.

For convenience, in the following we write $b^*, \hat{b}$ to mean $b^*(\hat{\tau}_h), \hat{b}(\hat{\tau}_h)$, and similarly $v^*, \hat{v}$ to mean $v^*(\hat{\tau}_h), \hat{v}(\hat{\tau}_h)$. $H$ is the set of exposures. Therefore, putting these together by a union bound,

$$
\begin{aligned}
\mathbb{P}\{\hat{h}_n \neq h_n^*\} &\leq \sum_h \mathbb{P}\{|\hat{M}_n(h) - M_n^*(h)| > \Delta_n/2\} \\
&\leq |H|\mathbb{P}\{|\hat{b}^2 + \hat{v} - b^{*2} - v^*| > \Delta_n/2\} \\
&\leq |H|\left(\mathbb{P}\{|\hat{b}^2 - b^{*2}| > \Delta_n/4\} + \mathbb{P}\{|\hat{v} - v^*| > \Delta_n/4\}\right) \\
&\leq |H|\left(\mathbb{P}\{(\hat{b} - b^*)^2 > \Delta_n/8\} + \mathbb{P}\{(\hat{b} - b^*)^2 > \Delta_n^2/(16U_n)^2\}\right. \\
&\quad \left. + \mathbb{P}\{|\hat{v} - v^*| > \Delta_n/4\}\right) \\
&\leq |H|\left(\mathbb{P}\{(\hat{\gamma}_n - \gamma_n^*)^2 > \Delta_n/8\} + \mathbb{P}\{(\hat{\gamma}_n - \gamma_n^*)^2 > \Delta_n^2/(16U_n)^2\}\right. \\
&\quad \left. + \mathbb{P}\{|\hat{v} - v^*| > \Delta_n/4\}\right) \\
&\leq 3|H|\max\left\{\exp\left(-\frac{\Delta_n np(1-p)}{8cd_{\max}} + 1\right),\right. \\
&\quad \exp\left(-\frac{\Delta_n^2 np(1-p)}{(16U_n)^2 cd_{\max}} + 1\right), \\
&\quad \left. 6\exp\left(-\frac{Cn\left(\frac{\Delta_n}{4} - \frac{cd_{\max}^2}{n}\right)}{\sqrt{A_{n,2}} + \sqrt{\left(\frac{\Delta_n}{4} - \frac{cd_{\max}^2}{n}\right)M_{n,2}}}\right)\right\}.
\end{aligned}
$$

The proof for the cluster-level Bernoulli randomization follows immediately.

$\square$

### A.5.2  Proof to Corollary 4.6

*Proof of Corollary 4.6.* Suppose $\sup_{e_i} |f(e_i) - \gamma^*| \leq \delta$. Let $\tilde{\delta} := 16\delta^2 + 8\delta U_n^*$. We begin by considering the difference between the MSE under the true best average linear fit and the true MSE:

$$
\begin{aligned}
|M_n^*(h) - M_n^{**}(h)| &= |b_n^{*2}(h) - b_n^{**2}(h)| \\
&\leq |(b_n^*(h) - b_n^{**}(h))(b_n^*(h) - b_n^{**}(h))| \\
&\leq |(b_n^*(h) - b_n^{**}(h))((b_n^*(h) - b_n^{**}(h)) + 2b_n^{**}(h))| \\
&\leq \left((b_n^*(h) - b_n^{**}(h))\right)^2 + 2|b_n^*(h) - b_n^{**}(h)|U_n^* \\
&\leq (4\delta)^2 + 2U_n^*(4\delta) \equiv \tilde{\delta}.
\end{aligned}
$$

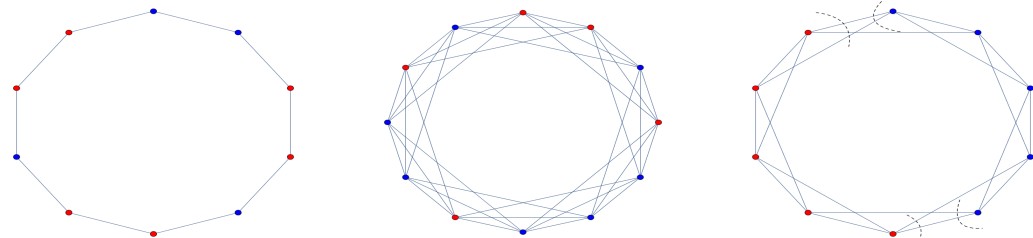

Figure 5: Circulant graphs with unit-level randomization and cluster-level randomization. Blue and red nodes represent treated and control units, respectively. Left: (1st-power) cycle graph with Ber(0.5) unit randomization. Center: 3rd-power cycle graph with Bernoulli(0.5) unit randomization. Right: 2nd-power cycle graph with Bernoulli(0.5) cluster randomization, with clusters of size 5 (=2k + 1).

This gives us,

$$
\begin{aligned}
&\mathbb{P}\{\hat{h}_n \neq \hat{h}_n^{**}\} \\
&\leq \sum_{h \in H} \mathbb{P}\{|\hat{M}_n(h) - M_n^{**}(h)| > \Delta_n/2\} \\
&= \sum_{h \in H} \mathbb{P}\{|\hat{M}_n(h) - M_n^*(h) + M_n^*(h) - M_n^{**}(h)| > \Delta_n/2\} \\
&\leq \sum_{h \in H} \mathbb{P}\{|\hat{M}_n(h) - M_n^*(h)| + |M_n^*(h) - M_n^{**}(h)| > \Delta_n/2\} \\
&\leq \sum_{h \in H} \mathbb{P}\{|\hat{M}_n(h) - M_n^*(h)| > \Delta_n/2 - \tilde{\delta}\}
\end{aligned}
$$

where the second inequality is obtained by applying a triangle inequality.

Finally, applying Theorem 4.5 gives us the result. $\qquad\square$

### A.6 Toy examples: Tradeoffs in Circulant graphs

We consider unit-level and cluster-level Bernoulli($p$) randomizations in the $k$th-power cycle graphs, see Figure 5. We say a graph is a $k$th-power cycle graph if there exists an edge between each node and $2k$ of its nearest neighbors [Ugander et al., 2013].

**Proposition A.1** (Absolute bias in k-th power cycle graphs under unit-randomization). *When $p = 1/2$ and the potential outcome model is simply linear, i.e. $Y_i = \alpha + \beta z_i + \gamma e_i$, the absolute bias of the Horvitz-Thompson estimator for a given threshold $h \equiv l/2k$ for $l = 0, 2, ..., 2k$, in the kth-power cycle graph, under unit-level randomization, is equal to $\gamma \times \left[ \frac{\sum_{r=l}^{2k} (r/k - 1)\binom{2k}{r}}{\sum_{r=l}^{2k} \binom{2k}{r}} - 1 \right].$*

**Proposition A.2** (Variance in k-th power cycle graphs under unit-randomization). *Denote the degree of the nodes by $d(= 2k)$. When $p = 1/2$ and the potential outcome model is simply linear, i.e. $Y_i = \alpha + \beta z_i + \gamma e_i$, the variance of the Horvitz-Thompson estimator for a given threshold h, in the k-th power cycle graph, under unit-level randomization is proportional to*

$$
\mathcal{O}\left( \frac{1}{np^{dh}} \left[ (\alpha + \beta + \gamma dh)^2 + (\alpha + \beta + \gamma d(1-h))^2 - 2\gamma h(1-h)d \right] \right).
$$

Therefore, the optimal threshold $h^*$ here depends on $\gamma, \beta, n, p, d$, the graph structure, and the number of other candidate thresholds.

From this toy example, it is not difficult to see that for unit-level Bernoulli randomization for more general graphs, the squared-bias would be $\Theta(\gamma^2 h^2)$, and the variance would be $\Theta(p^{-dh}\beta^2/n)$. Therefore, minimizing the MSE corresponds to balancing these quantities.

For $k$th-power cycle graphs, we also consider cluster-randomized design with cluster size $2k + 1$. In [Ugander et al., 2013], the authors show that this clustering size minimizes variance under full neighborhood exposure. We state the approximate absolute bias and variance for this setting in the following propositions. Here, define the threshold $h = \frac{l}{d}$ with $l = 0, 1, ..., d$.

**Proposition A.3** (Absolute bias in k-th power cycle graph under cluster-randomization). *When $p = 1/2$, and the potential outcome model is simply linear, i.e. $Y_i = \alpha + \beta z_i + \gamma e_i$, the bias of the Horvitz-Thompson estimator for a given threshold $h$ in the k-th power cycle graph under cluster randomization, with cluster-sizes $2k + 1$, is approximately $2\gamma(h - 1)$ for $h \geq 1/2$.*

**Proposition A.4** (Variance in the k-th power cycle graph under cluster-randomization). *When $p = 1/2$ and the potential outcome model is simply linear, i.e. $Y_i = \alpha + \beta z_i + \gamma e_i$, the variance of the Horvitz-Thompson estimator for a given threshold $h$, in the k-th power cycle graph under cluster-level randomization, with cluster-sizes $2k + 1$, is proportional to*

$$\frac{1}{np^2}(3d + 1 - 2dh)[(\beta + \gamma h)^2 + (\gamma(1 - h))^2] = \Theta(\frac{\beta^2 h^2}{np^{2h}}).$$

Under cluster randomization with cluster sizes $2k + 1$ for the k-th power cycle graphs, the variance grows linearly in the degrees of the graph. Therefore, informally, compared to the unit-randomized design setting, higher node degrees lead to stronger bias than variance for a fixed exposure function $\psi(\cdot, e_i)$.

### A.6.1 Proofs to Propositions A.1, A.2, A.3, A.4

*Proof to Proposition A.1.* When $p = 1/2$ and the potential outcome model is simply linear, i.e. $Y_i = \alpha + \beta z_i + \gamma e_i$, the absolute bias of the Horvitz-Thompson estimator for a given threshold $h \equiv l/2k$ for $l = 0, 2, ..., 2k$, in the $k$th-power cycle graph under unit-randomization is equal to

$$\frac{1}{\mathbb{P}\{z_i = 1, e_i \geq h\}} \sum_{x_i \in X} \frac{\mathbf{1}\{x_i \geq h\}}{|x_i : x_i \in X_i \cap x_i \geq h|} \mathbb{E}[\mathbf{1}\{z_i = 1, e_i = x_i\}]y_i(x_i)$$

$$- \frac{1}{\mathbb{P}\{z_i = 0, e_i \leq 1 - h\}} \sum_{x_i \in X} \frac{\mathbf{1}\{x_i \leq 1 - h\}}{|\{x_i : x_i \in X_i \cap x_i \leq 1 - h\}|} \mathbb{E}[\mathbf{1}\{z_i = 0, e_i = x_i\}]y_i(x_i)$$

$$- (\beta + \gamma)$$

$$= \frac{1}{\sum_{r=l}^{2k} \binom{2k}{r} p^{2r}} \sum_{x_i \in X} \frac{\mathbf{1}\{x_i \geq h\}}{|\{x_i : x_i \in X_i \cap x_i \leq 1 - h\}|} \mathbb{E}[\mathbf{1}\{z_i = 1, e_i = x_i\}]y_i(x_i)$$

$$- \frac{1}{\sum_{r=l}^{2k} \binom{2k}{r} p^{2r}} \sum_{x_i \in X} \frac{\mathbf{1}\{x_i \leq 1 - h\}}{|\{x_i : x_i \in X_i \cap x_i \leq 1 - h\}|} \mathbb{E}[\mathbf{1}\{z_i = 0, e_i = x_i\}]y_i(x_i)$$

$$- (\beta + \gamma)$$

$$= \frac{1}{\sum_{r=l}^{2k} \binom{2k}{r}} \left( \sum_{r=l}^{2k} \binom{2k}{r}\beta + \binom{2k}{r}(r/k - 1)\gamma \right) - (\beta + \gamma)$$

$$= \frac{1}{\sum_{r=l}^{2k} \binom{2k}{r}} \left( \sum_{r=l}^{2k} \binom{2k}{r}(r/k - 1)\gamma \right) - \gamma$$

where $X = \{0, 1/2k, 2/2k, ..., 1\}$. $\qquad\square$

*Proof to Proposition A.2.* When $p = 1/2$ and the potential outcome model is simply linear, i.e. $Y_i = \alpha + \beta z_i + \gamma e_i$, it is not difficult to see that the variance of the Horvitz-Thompson estimator for a given threshold $h \equiv l/2k$, in the $k$-th power cycle graph under unit-level randomization is proportional to

$$\frac{1}{n} \sum_{l=0}^{2k} \binom{2k}{l} \mathbf{1}\{l/2k \geq h\} \left[ (\alpha + \beta + \gamma l/2k)^2 (\frac{1}{\sum_{l=0}^{2k} \binom{2k}{l} \mathbf{1}\{l/2k \geq h\} p^{2l}} - 1) \right.$$

$$+ 2(\beta + \gamma l/(2k))(\gamma(1 - l/2k))(2k + 1 - \frac{2k}{\sum_{l=0}^{2k} \binom{2k}{l} \mathbf{1}\{l/2k \geq h\} p^{2l}})$$

$$\left. + (\alpha + \gamma(1 - l/2k))^2 (\frac{1}{\sum_{l=0}^{2k} \binom{2k}{l} \mathbf{1}\{l/2k \geq h\} p^{2l}} - 1) \right].$$

Considering the dominating terms gives us the result. $\qquad\square$

*Proof to Proposition A.3.* When $p = 1/2$ and the potential outcome model is simply linear, i.e. $Y_i = \alpha + \beta z_i + \gamma e_i$, the absolute bias of the Horvitz-Thompson estimator for a given threshold $h \equiv l/2k$ for $l = k, k+1, ..., 2k$,

in the $k$th-power cycle graph under cluster-randomization is equal to is equal to

$$\frac{1}{\mathbb{P}\{z_i = 1, e_i \geq h\}} \sum_{x_i \in X} \frac{\mathbf{1}\{x_i \geq h\}}{|x_i : x_i \in X_i \cap x_i \geq h|} \mathbb{E}[\mathbf{1}\{z_i = 1, e_i = x_i\}] y_i(x_i)$$

$$- \frac{1}{\mathbb{P}\{z_i = 0, e_i \leq 1 - h\}} \sum_{x_i \in X} \frac{\mathbf{1}\{x_i \leq 1 - h\}}{|x_i : x_i \in X_i \cap x_i \leq 1 - h|} \mathbb{E}[\mathbf{1}\{z_i = 0, e_i = x_i\}] y_i(x_i)$$

$$- (\beta + \gamma)$$

$$= \frac{(p/(2k+1) + 2k/(2k+1) \cdot p^2)(\beta + \gamma) + 2p/(2k+1) \sum_{r=1}^{k} \mathbf{1}\{r \geq d - l\}(\beta + \gamma \cdot (2l-d)/d)}{p/(2k+1) + p^2 \cdot 2k/(2k+1) + 2p/(2k+1) \sum_{r=1}^{k} \mathbf{1}\{r \geq d - l\}}$$

$$- (\beta + \gamma)$$

$$= \mathcal{O}(2\gamma(h-2))$$

where $X = \{0, 1/2k, 2/2k, ..., 1\}$. When $l < k$, the bias scales is the same as when $l = k$. $\square$

*Proof to Proposition A.4.* When $p = 1/2$ and the potential outcome model is simply linear, i.e. $Y_i = \alpha + \beta z_i + \gamma e_i$, it is not difficult to see that the variance of the Horvitz-Thompson estimator for a given threshold $h \equiv l/2k$, in the $k$-th power cycle graph under cluster-randomization, with cluster-size $2k + 1$, is proportional to

$$\sum_{u=0}^{d} \mathbf{1}\{u/d \geq h\} \binom{d}{u} \sum_{i=1}^{n} (\frac{(\beta + \gamma u/d)^2}{n^2})[(3d + 1 - 2l)(\frac{1}{p^2} - 1) + (2l + 1)(\frac{1}{p} - 1)]$$

$$+ \sum_{u=0}^{d} \mathbf{1}\{u/d \geq h\} \binom{d}{u} \sum_{i=1}^{n} (\frac{(\gamma(\frac{d-u}{d}))^2}{n^2})[(3d + 1 - 2l)(\frac{1}{p^2} - 1) + (2l + 1)(\frac{1}{p} - 1)]$$

$$- \sum_{u=0}^{d} \mathbf{1}\{u/d \geq h\} \binom{d}{u} \frac{2}{n}(\beta + \gamma \cdot u/d)(\gamma \cdot \frac{d-u}{d}).$$

Considering the dominating terms gives us the result. $\square$

## A.7 Simulations

### A.7.1 Experimental details and results on the Amazon product similarity graph data

We also evaluate the performance of our estimator on the Amazon (DVD) products similarity network [Leskovec et al., 2007]. We ran experiments with synthetic potential outcomes, averaging over 1000 trials, under unit-level randomization and spectral cluster-level randomizations. We generate simulated data using the linear model with $\psi(z_i, e_i) = g(z_i) + f(e_i)$, $\alpha_i = 10$, $g(z_i) = \beta z_i = 10 z_i$, $f(e_i) = \gamma e_i$, and $\epsilon_i$ is generated from $\mathcal{N}(0, 1)$ under a unit-level Bernoulli(0.5) randomization design. We focus on varying the ratio $\gamma/\beta$ as we consider a fixed graph. Figure 6 shows that we consistently perform better than existing fixed estimators.

The simulations were run on a CPU. Our experiments focus on a subset of (the first) 1000 nodes of the 19828-node DVD graph. In particular, we considered the 17924-node subgraph by removing all isolated nodes. To compute the exposure probabilities, we used $10^6$ simulations. We then selected the first 1000 nodes of the 19828 to analyze. 200 replicates were run, generating random treatment assignments and the corresponding outcomes. For each of the replicates, 1000 separate simulation runs were generated to compute the oracle MSEs. In total, this took approximately 2 hours. The graph data is available at: https://snap.stanford.edu/data/amazon-meta.html

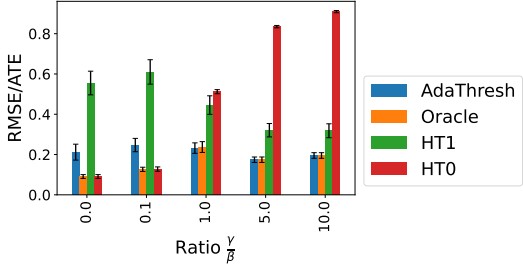

Figure 6: RMSE (normalized by the ATE) across different thresholds on the Amazon product network data. The error bars are two times the standard deviation.

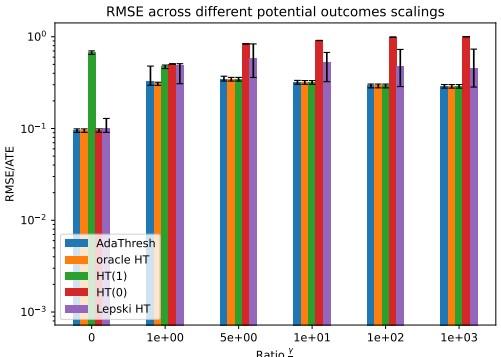
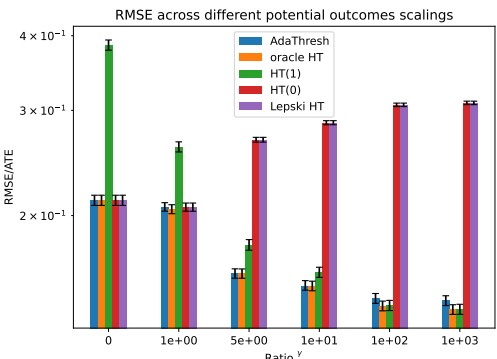

Figure 7: RMSE (normalized by the ATE) of different Horvitz-Thompson estimators. Left: 2nd-power cycle graph under unit-level Ber(0.5) randomization. Right: 2nd-power cycle graph under cluster-level Ber(0.5) randomization with cluster sizes 5 ($=2k+1$). The error bars are two times the standard deviation.

### A.7.2 Experimental details on simulated graphs

All synthetic graph simulations were run on a machine of Intel® Xeon® processors with 48 CPU cores, and 50GB of RAM. We simulated 1000 replicates, generating random treatment assignments and the corresponding outcomes, with each oracle MSE computed using 1000 separate simulation runs. The exposure probabilities under each threshold were computed as proposed in [Aronow and Samii, 2017] using 20000 simulation iterations. In total, this took approximately 30 minutes on average (across the different potential outcomes, and graph settings). The code is available at: https://github.com/Vydhourie/AdaThresh.git

In the following, we focus on key experimental results on circulant graphs (see Section A.6 for discussions on circulant graphs).

### A.7.3 More Simulations for Horvitz-Thompson estimator with adaptive exposure thresholds

In Figure 7, we simulate outcomes from the linear model with $\psi(z_i, e_i) = g(z_i) + f(e_i)$, $\alpha_i = 10$, $g(z_i) = \beta z_i = 10 z_i$, $f(e_i) = \gamma e_i$, and fixed $\epsilon_i$ generated from $\mathcal{N}(0,1)$ for a 1000-node 2nd-power cycle graph.

### A.7.4 Simulations for non-linear potential outcomes models

We investigate the robustness of our estimator to potential outcomes models that are non-linear in the exposure. In particular, we consider simulations from the sigmoid, and sine exposure functions. In Figure 8, we display the performance of our estimator under a sigmoid (left) and sine (right) interference function, respectively. Our adaptive threshold Horvitz-Thompson estimator (AdaThresh) generally improves upon other existing Horvitz-Thompson estimators.

### A.7.5 Simulations for Difference-in-Means estimator with adaptive exposure thresholds

We consider our approach using the Difference-in-Means estimator incorporating exposure thresholds:

$$\hat{\tau}_{\text{DiM}_h} = \frac{\sum_{i=1}^n \mathbf{1}\{z_i = 1, e_i \geq h\} Y_i}{\sum_{i=1}^n \mathbf{1}\{z_i = 1, e_i \geq h\}} - \frac{\sum_{i=1}^n \mathbf{1}\{z_i = 0, e_i \leq 1 - h\} Y_i}{\sum_{i=1}^n \mathbf{1}\{z_i = 0, e_i \leq 1 - h\}}. \tag{14}$$

We use the following bias estimator:

$$\hat{b}(\hat{\tau}_h) = \sum_{i=1}^n \frac{(1 - e_i)\hat{\gamma}_n \mathbf{1}\{z_i = 1, e_i \geq h\}}{\sum_{i=1}^n \mathbf{1}\{z_i = 1, e_i \geq h\}} + \sum_{i=1}^n \frac{e_i \hat{\gamma}_n \mathbf{1}\{z_i = 0, e_i \leq 1 - h\}}{\sum_{i=1}^n \mathbf{1}\{z_i = 0, e_i \leq 1 - h\}},$$

where $\hat{\gamma}$ is the linear regression coefficient for the exposure variable.

We use the following variance estimator, decomposing it into its treatment and control parts:

$$\hat{v}(\hat{\tau}) = \frac{2}{n-1}\left(\hat{v}_T + \hat{v}_C\right)$$

where

$$\hat{v}_T = \frac{1}{n_1} \sum_{i=1}^n \left(Y_i \mathbf{1}\{z_i = 1, e_i \geq h\} - \frac{1}{n_1} \sum_{i=1}^n Y_i \mathbf{1}\{z_i = 1, e_i \geq h\}\right)^2$$

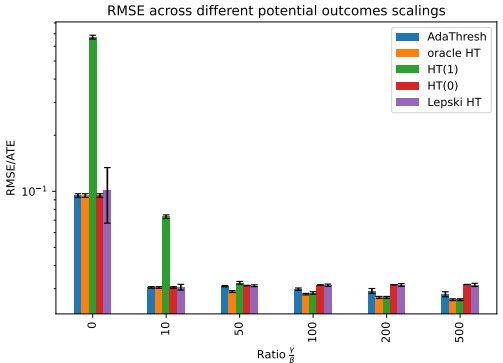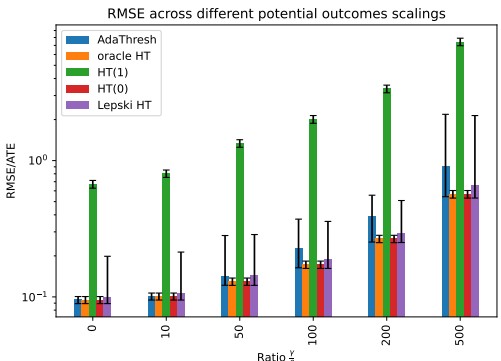

Figure 8: RMSE (normalized by the ATE) induced by the different Horvitz-Thompson estimators. We take $\psi(z_i, e_i) = g(z_i) + f(e_i)$. Left: 2nd-power cycle graph under unit-level Ber($p$), $p = 0.5$, randomization under sigmoid $f(e_i) = \gamma/(1 + \exp(-10(e_i - p)))$. Right: 2nd-power cycle graph under unit-level Ber($p$), $p = 0.5$, randomization with $f(e_i) = \gamma(1 - \sin(\pi \cdot e_i))$ being a sine function. We focus on varying the ratio $\gamma/\beta$ as we consider a fixed graph, with $n = 1000$. The error bars are two times the standard deviation.

where $n_1 := \sum_{i=1}^{n} \mathbf{1}\{z_i = 1, e_i \geq h\}$, and

$$\hat{v}_C = \frac{1}{n_0} \sum_{i=1}^{n} \left( Y_i \mathbf{1}\{z_i = 0, e_i \leq 1 - h\} - \frac{1}{n_0} \sum_{i=1}^{n} Y_i \mathbf{1}\{z_i = 0, e_i \leq 1 - h\} \right)^2$$

where $n_0 := \sum_{i=1}^{n} \mathbf{1}\{z_i = 0, e_i \leq 1 - h\}$.

We compare the performance of our adaptive estimator to the vanilla difference-in-means estimator, the difference-in-means analogue of the vanilla Horvitz-Thompson estimator, and the difference-in-means estimator with a threshold plugin via Lepski's method. We write these out below:

- vanilla difference-in-means estimator

$$\hat{\tau}_{\text{DiM}_0} = \frac{\sum_{i=1}^{n} \mathbf{1}\{z_i = 1\} Y_i}{\sum_{i=1}^{n} \mathbf{1}\{z_i = 1\}} - \frac{\sum_{i=1}^{n} \mathbf{1}\{z_i = 0\} Y_i}{\sum_{i=1}^{n} \mathbf{1}\{z_i = 0\}} \tag{15}$$

- difference-in-means estimator at threshold $h = 1$

$$\hat{\tau}_{\text{DiM}_1} = \frac{\sum_{i=1}^{n} \mathbf{1}\{z_i = 1, e_i = 1\} Y_i}{\sum_{i=1}^{n} \mathbf{1}\{z_i = 1, e_i = 1\}} - \frac{\sum_{i=1}^{n} \mathbf{1}\{z_i = 0, e_i = 0\} Y_i}{\sum_{i=1}^{n} \mathbf{1}\{z_i = 0, e_i = 0\}} \tag{16}$$

- difference-in-means estimator at threshold $\hat{h}_{\text{Lepski}}$ where

$$\hat{h}_{\text{Lepski}} := \min\{h \in H : \cap_{h' \in H : h' \geq h} I(h') \neq \emptyset\}, \tag{17}$$

with

$$I(h) := [\hat{\tau}_{\text{DIM}_h} - 2\widehat{\text{SDEV}}(\hat{\tau}_{\text{DIM}_h}), \hat{\tau}_{\text{DIM}_h} + 2\widehat{\text{SDEV}}(\hat{\tau}_{\text{DIM}_h})], \tag{18}$$

and

$$\hat{\tau}_{\text{LepskiDiM}} = \sum_{i=1}^{n} \frac{\mathbf{1}\{z_i = 1, e_i \geq \hat{h}_{\text{Lepski}}\}}{\sum_{i=1}^{n} \mathbf{1}\{z_i = 1, e_i \geq \hat{h}_{\text{Lepski}}\}} Y_i - \sum_{i=1}^{n} \frac{\mathbf{1}\{z_i = 0, e_i \leq 1 - \hat{h}_{\text{Lepski}}\}}{\sum_{i=1}^{n} \mathbf{1}\{z_i = 0, e_i \leq 1 - \hat{h}_{\text{Lepski}}\}} Y_i \tag{19}$$

In Figures 9 and 10, we display the performance of our adaptive threshold (AdaThresh) Difference-in-Means estimators in comparison to other estimators. AdaThresh improves upon fixed threshold Difference-in-Means estimators.

### A.7.6 Simulations using local linear regression

We extend our global linear regression approach to a local linear regression one to estimate the rate of change of the bias. We write the new bias estimators for the Horvitz-Thompson and Difference-in-Means estimators. We illustrate the performance of the local linear regression in the settings above, as well as settings that significantly

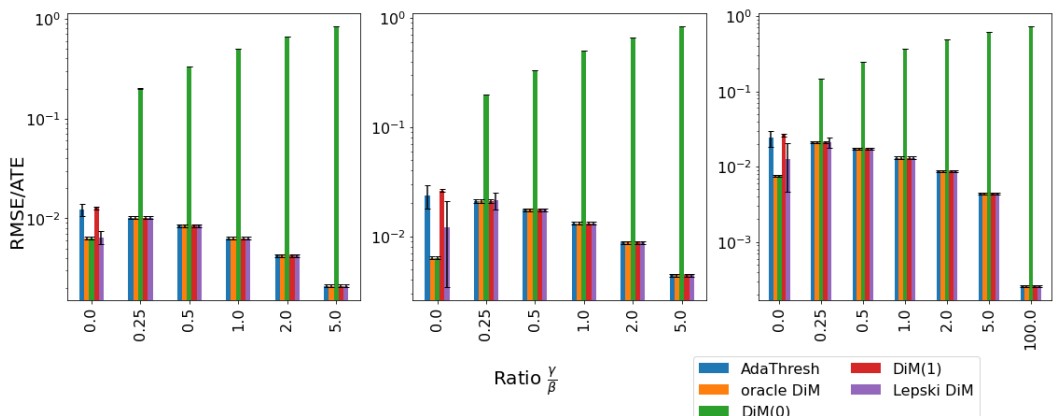

Figure 9: RMSE (normalized by the ATE) under the linear model with $\psi(z_i, e_i) = g(z_i) + f(e_i)$, $\alpha_i = 10$, $g(z_i) = \beta z_i = 10 z_i$, $f(e_i) = \gamma e_i$, and fixed $\epsilon_i$ generated from $\mathcal{N}(0, 1)$, induced by the different Difference-in-Means estimators. We focus on varying the ratio $\gamma/\beta$ as we consider a fixed graph. Left: Cycle graph under unit-level Ber(0.5) randomization. Center: 2nd-power cycle graph under unit-level Ber(0.5) randomization. Right: 2nd-power cycle graph under cluster-level Ber(0.5) randomization with cluster sizes 5 (=2k + 1). The error bars are two times the standard deviation.

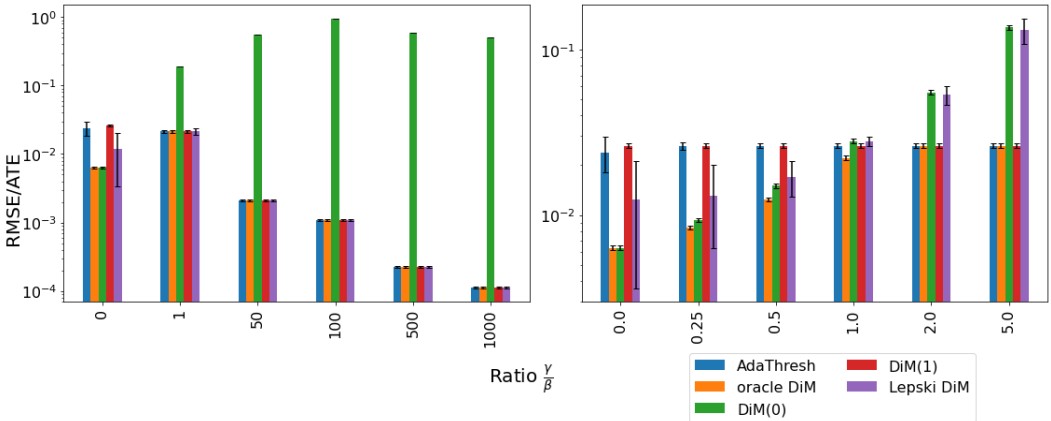

Figure 10: RMSE (normalized by the ATE) induced by the different Difference-in-Means estimators. We take $\psi(z_i, e_i) = g(z_i) + f(e_i)$. Left: 2nd-power cycle graph under unit-level Ber(p), $p = 0.5$, randomization under sigmoid $f(e_i) = \gamma/(1 + \exp(-e_i))$. Right: 2nd-power cycle graph under unit-level Ber(p), $p = 0.5$, randomization with $f(e_i) = \gamma(1 - \sin(\pi \cdot e_i))$ being a sine function. We focus on varying the ratio $\gamma/\beta$ as we consider a fixed graph, with $n = 1000$. The error bars are two times the standard deviation.

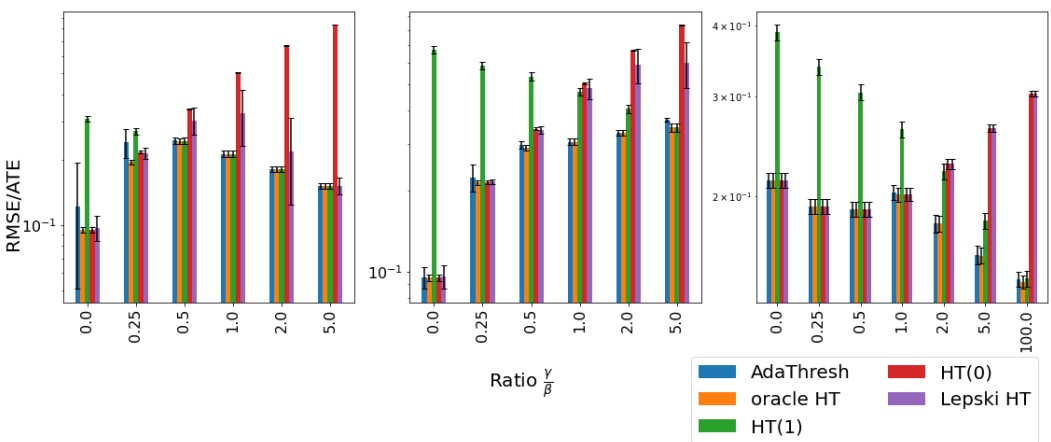

Figure 11: RMSE (normalized by the ATE) under the linear model with $\psi(z_i, e_i) = g(z_i) + f(e_i)$, $\alpha_i = 10$, $g(z_i) = \beta z_i = 10z_i$, $f(e_i) = \gamma e_i$, and fixed $\epsilon_i$ generated from $\mathcal{N}(0, 1)$, induced by the different (local) Horvitz-Thompson estimators. We focus on varying the ratio $\gamma/\beta$ as we consider a fixed graph, with $n = 1000$. Left: Cycle graph under unit-level Ber(0.5) randomization. Center: 2nd-power cycle graph under unit-level Ber(0.5) randomization. Right: 2nd-power cycle graph under cluster-level Ber(0.5) randomization with cluster sizes 5 (=2k + 1). The error bars are two times the standard deviation.

deviates from linearity. We note that local and global linear regression involve distinct bias-variance trade-offs, which we do not pursue here as they fall outside the scope of the paper.

We write the bias estimator for the Horvitz-Thompson estimator as:

$$\hat{b}(\hat{\tau}_h) = \frac{1}{n} \sum_{i=1}^{n} \frac{(1 - e_i)\hat{\gamma}_n^{(h)} \mathbf{1}\{z_i = 1, e_i \geq h\}}{\mathbb{P}\{z_i = 1, e_i \geq h\}} + \frac{1}{n} \sum_{i=1}^{n} \frac{e_i \hat{\gamma}_n^{(1-h)} \mathbf{1}\{z_i = 0, e_i \leq 1 - h\}}{\mathbb{P}\{z_i = 0, e_i \leq 1 - h\}},$$

where $\hat{\gamma}_n^{(h)}$ and $\hat{\gamma}_n^{(1-h)}$ are the local linear regression coefficients for the exposure variable in the intervals $[h, 1]$, and $[0, 1 - h]$, respectively.

Similarly, write the bias estimator for the Difference-in-Means estimator as:

$$\hat{b}(\hat{\tau}_h) = \sum_{i=1}^{n} \frac{(1 - e_i)\hat{\gamma}_n^{(h)} \mathbf{1}\{z_i = 1, e_i \geq h\}}{\sum_{i=1}^{n} \mathbf{1}\{z_i = 1, e_i \geq h\}} + \sum_{i=1}^{n} \frac{e_i \hat{\gamma}_n^{(1-h)} \mathbf{1}\{z_i = 0, e_i \leq 1 - h\}}{\sum_{i=1}^{n} \mathbf{1}\{z_i = 0, e_i \leq 1 - h\}},$$

where $\hat{\gamma}_n^{(h)}$ and $\hat{\gamma}_n^{(1-h)}$ are the local linear regression coefficients for the exposure variable in the intervals $[h, 1]$, and $[0, 1 - h]$, respectively.

Figures 11, 13, 12, and 14, display how local linear regression bias estimates perform.

## A.8 Discussion on relaxing the bounded degree assumption

In more general settings, we can relax Assumption 4.3 by partitioning the exposures into exposure bins $E_b$, $b = 1, 2, \ldots, K$, each associated with an "effective exposure" $\tilde{e}_b$. That is, $[0, 1] = E_1 \cup E_2 \cup \ldots \cup E_K$, with $E_i \cap E_j = \emptyset$, for all $i \neq j \in [K]$. In particular, in the weighted graph setting, we can take the effective exposure of unit $i$, $\tilde{e}_i = \tilde{e}_b$ if and only if $i \in E_b$, and such that exposure positivity, for all $i \in [n]$, $r \in \{0, 1\}$, and $s \in [0, 1]$, $\mathbb{P}(z_i = r, \tilde{e}_i = s) > 0$, is satisfied. For instance, for uniformly sized bins with associated effective exposures $\tilde{e}_b$ that well approximate $e_i$, $|E_b| = 1/K$. Let $B_r(i)$ denote the ball of radius $r$ around unit $i$, such that it decomposes, $|B_1(i)| = |B_1^S(i)| + |B_1^W(i)|$ for all $i \in [n]$. $B_1^S(i)$ is the ball of radius 1 around unit $i$ with strong connections $\eta_i^s$, and $|B_1^W(i)|$ be the ball of radius 1 around unit $i$ with weak connections $\delta_i^w$ such that $\sup_i \frac{\eta_i^s}{\delta_i^w} \to \infty$. We choose our partition $B_1, \ldots B_K$, for some $K$, as a function of $\delta_i^w$, and $B_1^W(i)$, $i \in [n]$, such that the exposure bins well approximate the true exposures. Additionally, for all $i \in [n]$, we require that $|B_1^S(i)| = \mathcal{O}(1)$. With an abuse of notation, write $e_i(\mathcal{N})$ to mean the exposure of unit $i$ as a function of the treated units in the subset $\mathcal{N} \subseteq B_1(i)$. Then, we require also that $e_i(B_1^S(i)) = \Omega(e_i(B_1(i)))$. Additionally, $B_1^S(i)$ must satisfy the three conditions around "affinity sets" in [Chandrasekhar et al., 2023] generalizing [Aronow and Samii, 2017], with an additional condition that the sum of the affinity set covariances is $\Omega(n)$, in the bias terms as well as the variance terms in the MSE. This gives us uniform control on the estimated MSE

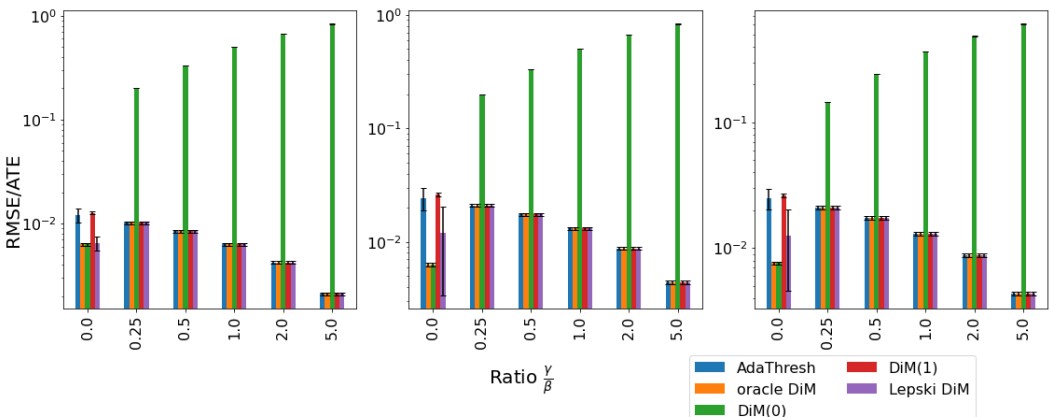

Figure 12: RMSE (normalized by the ATE) under the linear model with $\psi(z_i, e_i) = g(z_i) + f(e_i)$, $\alpha_i = 10$, $g(z_i) = \beta z_i = 10z_i$, $f(e_i) = \gamma e_i$, and fixed $\epsilon_i$ generated from $\mathcal{N}(0, 1)$, induced by the different (local) Difference-in-Means estimators. We focus on varying the ratio $\gamma/\beta$ as we consider a fixed graph, with $n = 1000$. Left: Cycle graph under unit-level Ber(0.5) randomization. Center: 2nd-power cycle graph under unit-level Ber(0.5) randomization. Right: 2nd-power cycle graph under cluster-level Ber(0.5) randomization with cluster sizes 5 (=2k + 1). The error bars are two times the standard deviation.

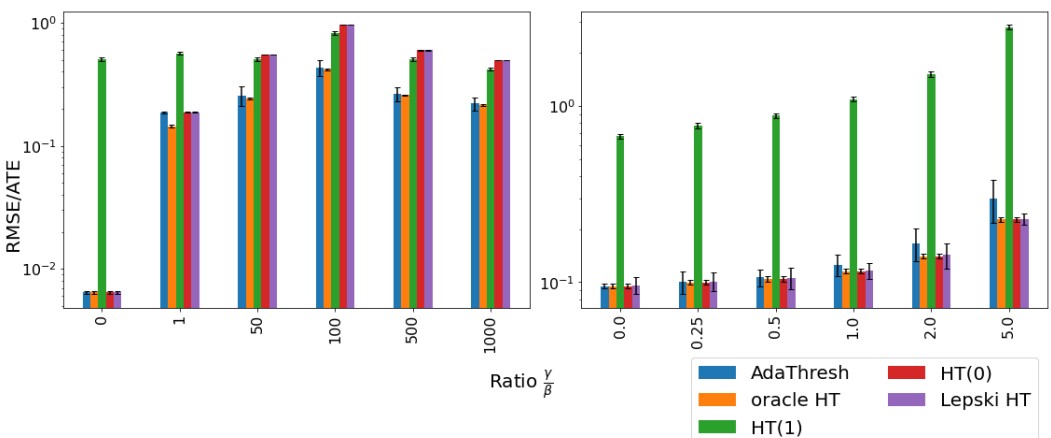

Figure 13: RMSE (normalized by the ATE) under the linear model with $\psi(z_i, e_i) = g(z_i) + f(e_i)$, $\alpha_i = 10$, $g(z_i) = \beta z_i = 10z_i$, and fixed $\epsilon_i$ generated from $\mathcal{N}(0, 1)$, induced by the different (local) Horvitz-Thompson estimators. Left: 2nd-power cycle graph under unit-level Ber(0.5) randomization under sigmoid $f(e_i) = \gamma/(1 + \exp{(-e_i)})$. Right: 2nd-power cycle graph under unit-level Ber(0.5) randomization with $f(e_i) = \gamma(1 - \sin(\pi \cdot e_i))$ being a sine function. We focus on varying the ratio $\gamma/\beta$ as we consider a fixed graph, with $n = 1000$. The error bars are two times the standard deviation.

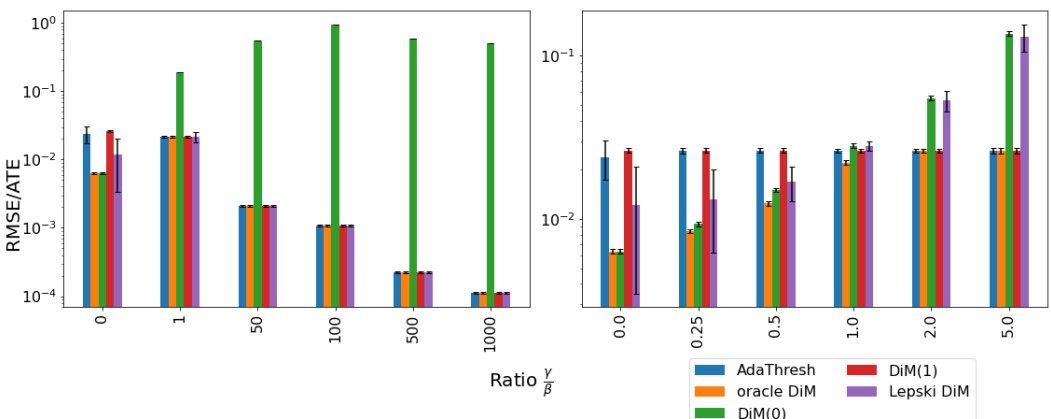

Figure 14: RMSE (normalized by the ATE) induced by the different (local) Difference-in-Means estimators. We take $\psi(z_i, e_i) = g(z_i) + f(e_i)$. Left: 2nd-power cycle graph under unit-level Ber(0.5) randomization under sigmoid $f(e_i) = \gamma/(1 + \exp(-e_i)))$. Right: 2nd-power cycle graph under unit-level Ber(0.5) randomization with $f(e_i) = \gamma(1 - \sin(\pi \cdot e_i))$ being a sine function. We focus on varying the ratio $\gamma/\beta$ as we consider a fixed graph, with $n = 1000$. The error bars are two times the standard deviation.

of the estimator with respect to the true MSE of the estimator and $E_b$, $b = 1, \ldots, K$, and all the results follow through. These conditions subsume the approximate neighborhood interference (ANI) condition in [Leung, 2022] analogously to $\alpha$-mixing conditions in spatial settings, such as in [Jenish and Prucha, 2009], when the maximum clique size in the network does not grow. Indeed, when the maximum clique size in the network does not grow, we can embed the $\psi$-dependent network in [Leung, 2022] in a lattice structure of a random field in a fixed-dimensional Euclidean space with $\alpha$-mixing just as in [Jenish and Prucha, 2009].

### A.9 Discussion on "double-dipping"

Since our approach involves "double-dipping" into the data, we need to make sure that there is no overfitting that occurs. The authors of [Chernozhukov et al., 2018], [Belloni et al., 2015], and [Kennedy, 2022] describe this in more detail. While sample-splitting would simply take care of this in the case of independent data, it does not apply to our setting where the data are dependent, as modeled by the network. One could potentially leverage results under the dependent-data setting, such as Hart and Vieu [1990], but this is outside the scope of our paper. We propose to use our approach on the whole sample data and argue this via empirical process theory arguments. In particular, we can think of our threshold $h$ as a nuisance parameter, and we show that our estimator for $h$ has a simple enough associated MSE function class. We note that it is sufficient to show that we are not overfitting by showing that the rate of convergence of the estimated MSE under $\hat{h}$ converges to the true MSE under the optimal threshold, under the best possible linear fit, $h^*$ at least at a $\mathcal{O}(n^{-1/2})$-rate.

**Proposition A.5.** *Suppose that Assumptions 2, 4.2, and 4.3 are satisfied. The corresponding bias terms in the estimated and true MSE, $\hat{M}_n^b$, and $M_n^b$, under "double prediction", satisfy*

$$\mathbb{E}\left[\sup_{\delta/2 < |h_n^* - \hat{h}_n| < \delta} \sqrt{n}\left(\hat{M}_n^b(\hat{h}_n) - M_n^b(\hat{h}_n) - \hat{M}_n^b(h^*) - M_n^b(h^*)\right)\right] \to 0,$$

*as $\delta \to 0$.*

Suppose that Assumptions 2, 4.2, and 4.3 are satisfied. We aim to show that the corresponding biases in the MSE terms under " double prediction" satisfy

$$\mathbb{E}\left[\sup_{\delta/2 < |h^* - h_n| < \delta} \sqrt{n}\left(\hat{M}_n^b(h_n) - M_n^b(h_n) - \hat{M}_n^b(h^*) - M_n^b(h^*)\right)\right] \to 0,$$

as $\delta \to 0$, where $x_i$ ranges over the set $X_i$ of possible fractions of degree $i$.

*Proof to Proposition A.5.* We begin by considering the empirical process in question. We write out

$$\hat{M}_n(h) = \left( \frac{1}{n} \sum_{i=1}^{n} \frac{\hat{\gamma}_h (1 - e_i) \mathbf{1}\{z_i = 1, e_i \geq h\}}{\mathbb{P}\{z_i = 1, e_i \geq h\}} + \frac{1}{n} \sum_{i=1}^{n} \frac{\hat{\gamma}_h (1 - e_i) \mathbf{1}\{z_i = 1, e_i \geq h\}}{\mathbb{P}\{z_i = 1, e_i \geq h\}} \right)^2$$

$$+ \sum_{i=1}^{n} \frac{\mathbf{1}\{z_i = 1, e_i \geq h\} Y_i^2}{n^2 \pi_i^1} \left( \frac{1}{\pi_i^1} - 1 \right)$$

$$+ \sum_{i=1}^{n} \sum_{\substack{j=1 \\ j \neq i}}^{n} \frac{\mathbf{1}\{z_i = 1, e_i \geq h\} \mathbf{1}\{z_j = 1, e_j \geq h\} Y_i Y_j}{n^2 \pi_{ij}^{11}} \left( \frac{\pi_{ij}^{11}}{\pi_i^1 \pi_j^1} - 1 \right)$$

$$+ \sum_{i=1}^{n} \frac{\mathbf{1}\{z_i = 0, e_i \leq 1 - h\} Y_i^2}{n^2 \pi_i^0} \left( \frac{1}{\pi_i^0} - 1 \right)$$

$$+ \sum_{i=1}^{n} \sum_{\substack{j=1 \\ j \neq i}}^{n} \frac{\mathbf{1}\{z_i = 0, e_i \leq 1 - h\} \mathbf{1}\{z_j = 0, e_j \leq 1 - h\} Y_i Y_j}{n^2 \pi_{ij}^{00}} \left( \frac{\pi_{ij}^{00}}{\pi_i^0 \pi_j^0} - 1 \right)$$

$$- \frac{2}{n^2} \sum_{i=1}^{n} \sum_{j \in [n]; \pi_{ij}^{10} > 0} \left( \mathbf{1}\{z_i = 1, e_i \geq h\} \mathbf{1}\{z_j = 0, e_j \leq 1 - h\} Y_i Y_j \right) \left( \frac{1}{\pi_i^1 \pi_j^0} - \frac{1}{\pi_{ij}^{10}} \right)$$

$$+ \frac{2}{n^2} \sum_{i=1}^{n} \sum_{j \in [n]; \pi_{ij}^{10} = 0} \left( \frac{\mathbf{1}\{z_i = 1, e_i \geq h\} Y_i^2}{2\pi_i^1} + \frac{\mathbf{1}\{z_j = 0, e_j \leq 1 - h\} Y_j^2}{2\pi_j^0} \right).$$

The absolute Horvitz-Thompson bias estimate at threshold $h$ (and at $1 - h$) is

$$\hat{b}(\hat{\tau}_h) = \frac{1}{n} \sum_{i=1}^{n} \frac{\hat{\gamma}_n (1 - e_i) \mathbf{1}\{z_i = 1, e_i \geq h\}}{\mathbb{P}(z_i = 1, e_i \geq h)} + \frac{1}{n} \sum_{i=1}^{n} \frac{\hat{\gamma}_n (e_i) \mathbf{1}\{z_i = 0, e_i \leq 1 - h\}}{\mathbb{P}(z_i = 0, e_i \leq 1 - h)} \tag{20}$$

Then, for every $h$, using the identity $x^2 - y^2 = (x + y)(x - y)$, and defining $U_n \in \mathbb{R}$ such that $U_n \geq M_n^b(h)$ for all $h$ (note that there exists such a $U_n < \infty$ by Assumptions 2, 4.2), we have

$$\hat{M}_n^b(\hat{\tau}_h) - M_n^b(\hat{\tau}_h) \leq 2U_n \left( \hat{b}(\hat{\tau}_h) - b^*(\hat{\tau}_h) \right) + \left( \hat{b}(\hat{\tau}_h) - b^*(\hat{\tau}_h) \right)^2,$$

where, from Section A.4, the difference between the bias estimate and the true bias induced by the best average linear fit is

$$\hat{b}(\hat{\tau}_h) - b^*(\hat{\tau}_h) = \left( \frac{1}{n} \sum_{i=1}^{n} \frac{\hat{\gamma}_n (1 - e_i) \mathbf{1}\{z_i = 1, e_i \geq h\}}{\mathbb{P}(z_i = 1, e_i \geq h)} - \frac{1}{n} \sum_{i=1}^{n} \sum_{x_i \in X_i} \frac{\mathbf{1}\{x_i \geq h\}}{|x_i : x_i \in X_i \cap x_i \geq h|} \gamma_n (1 - x_i) \right)$$

$$+ \left( \frac{1}{n} \sum_{i=1}^{n} \frac{\hat{\gamma}_n e_i \mathbf{1}\{z_i = 0, e_i \leq 1 - h\}}{\mathbb{P}(z_i = 0, e_i \leq 1 - h)} - \frac{1}{n} \sum_{i=1}^{n} \sum_{x_i \in X_i} \frac{\mathbf{1}\{x_i \leq 1 - h\}}{|x_i : x_i \in X_i \cap x_i \leq 1 - h|} \gamma_n (x_i) \right),$$

where $\gamma_n$ is the slope of the best average linear fit, and where $x_i$ ranges over the set $X_i$ of possible fractions of degree $i$.

We now proceed to consider each of the terms in $\hat{b}(\hat{\tau}_h) - b^*(\hat{\tau}_h)$ above. Each of the parentheses satisfies Donsker conditions. Indeed, under Assumptions 2, 4.2, and 4.3, they are sample-mean terms with uniformly bounded coefficients, and $\hat{\gamma}_n$ converges to $\gamma_n^*$ at a rate of $\mathcal{O}(n^{-1/2})$, which we compose with $3b^*(h)$ (see Section 3.4.3.2. in [Van Der Vaart et al., 1996]). Thus, the terms in $\hat{M}_n^b(h) - M_n^b(h)$ have bounded entropy integral. Under our bounded-variation potential-outcome model (Assumption 4.2), we have that $\log N_{[]}(\epsilon, \Theta_n^b, L_2(\mathbb{P}) \leq K \left( \frac{1}{\epsilon} \right)$, which gives us $\tilde{J}(\delta, \Theta_n^b, L_2(\mathbb{P})) \leq \delta^{1/2}$ [Van Der Vaart et al., 1996] (see also Example 19.11 in [Van der Vaart, 2000]).

Therefore, putting these together, we have the following maximal inequality for the bias terms

$$\mathbb{E} \left[ \sup_{\delta/2 < |h^* - \hat{h}_n| < \delta} \sqrt{n} \left( \hat{M}_n^b(h_n) - M_n^b(h_n) - (\hat{M}_n^b(h^*) - M_n^b(h^*)) \right) \right] \tag{21}$$

$$\leq \tilde{J}(\delta, \Theta_n^b, L_2(\mathbb{P})) \left( 1 + \frac{\tilde{J}(\delta, \Theta_n^b, L_2(\mathbb{P}))}{\delta \sqrt{n}} \right), \tag{22}$$

with $\tilde{J}(\delta, \Theta_n^b, L_2(\mathbb{P})) \leq \delta^{1/2}$.

$\square$

