# OpenReview forum: "Data-Adaptive Exposure Thresholds under Network Interference"
_NeurIPS.cc/2025/Conference — NeurIPS 2025 poster_

### Official Review · Reviewer_Y8AE · 2025-06-24

**Clarity:** 4
**Significance:** 3
**Originality:** 3
**Rating:** 5
**Confidence:** 4

**Summary:**

This paper considers estimate the average treatment effect with special attention to the scenario the SUTVA assumption does not hold. The paper considers exposure mapping and proposes a data-adaptive method to estimate the $h$-fractional threshold  for minimizing the Horvitz-Thompson estimator’s MSE in randomized controlled trials with interference, focusing on estimating $h$ with bias and variance tradeoff of the ATE.

**Questions:**

(1) Can it be generalized to nonlinear potential outcome models such as generalized linear models?
(2) The paper estimates the ATE of the form:
\begin{equation}
\tau = \frac{1}{n} \sum_{i = 1}^{n} y_i(1, 1) - \frac{1}{n} \sum_{i = 1}^{n} y_i(0, 0).
\end{equation}
Is it possible to estimate the effect of the treatment itself, excluding the influence of the exposure?

**Ethical Concerns:**

["NO or VERY MINOR ethics concerns only"]

**Final Justification:**

This paper proposes a data-adaptive method to select the h-fractional threshold for exposure mappings in randomized controlled trials with interference, aiming to minimize the MSE of the Horvitz-Thompson estimator by leveraging a first-order approximation, and demonstrates its superiority through extensive simulations. I decide to raise my score to 5.

**Limitations:**

Yes

**Quality:**

3

**Strengths And Weaknesses:**

Strengths：（1）Clear objective function for threshold \( h \) estimation, with rigorous bounds on estimation error.  （2） Experiments show strong real-data performance.

Weakness：Limited to linear potential-outcome models only.

---

> ### Author Rebuttal · Authors · 2025-07-27
>
> We thank Reviewer Y8AE for the thoughtful questions. We appreciate it, and would like to address the reviewer’s questions and concerns below, and hope that our response provides more clarity on the focus of our work.
>
> Response to Concern 1 / Question 1:
>
> We are not assuming a linear potential outcome model. Our model is much more general (please refer to eq. 1). We are simply using linear regression for a first-order approximation to the potential outcomes model for bias estimation. Only in the special case of a true linear model is this exact. We would also like to draw the reviewer’s attention to Baird et al.(2018) , where the authors leverage the slope of spillovers with respect to treatment saturation . Similar to our approach, this work uses the linear model, i.e. the slope, of the spillovers for an approximation without assuming it as the generating model. This is discussed in lines 151–157 of the paper.
>
> In Corollary 4.6, we provide error bounds for models of the form presented on line 112. To demonstrate the method’s robustness beyond linear settings, we report simulation results for nonlinear potential outcomes models in Section A.7.4. Finally, we extend our method to incorporate local regression approaches in Section A.7.6, and apply these to nonlinear outcome models as well. This is discussed in lines 301–313 of the paper.
>
> Question 2: “Is it possible to estimate the effect of the treatment itself, excluding the influence of the exposure?”..
> This is a great question. Extending our approach to other estimands, such as the direct effect, is indeed possible, though the characterization of bias and variance would be somewhat different. For example, to estimate the direct effect, one might focus on treated units whose neighbors are all assigned to control, yielding an unbiased estimate under certain conditions. In this case, rather than defining extremes as $(z_i=1,e_i \geq h)$ and $(z_i=0, e_i \leq 1−h)$, one might instead consider $(z_i=1,e_i \in x_h)$ and $(z_i=0,e_i \in x_h)$ for some interval $x_h=[x−h,x+h]$, where $x_h$ forms part of a cover of $[0,1]$. While the underlying principles remain similar, the bias–variance trade-offs do not translate in a straightforward manner between estimands, and would need to be re-analyzed in each case.
>
> We hope our response has provided more clarity on the motivations and the focus of our work. We thank the reviewer once again for the thoughtful questions, and we hope that this discussion has enhanced your perspective on our paper.

---

> > ### Comment · Reviewer_Y8AE · 2025-08-05
> > **Thank you**
> >
> > I’ve reviewed the entire paper, and it is indeed the case that your work does not assume a linear potential outcome model, as noted in line 152. Based on your rebuttal, I have clarified the key points of your research as follows.

---

### Official Review · Reviewer_mQMx · 2025-06-29

**Clarity:** 3
**Significance:** 3
**Originality:** 3
**Rating:** 5
**Confidence:** 2

**Summary:**

This paper considers the problem of average treatment effect (ATE) estimation under network interference.  In this setting, the treatment of neighbors can impact the outcome for an individual, and the ATE is defined as the average outcome when all units are treated (including the effect of neighbors on each other), versus the average outcome when no units are treated.  However, in a randomized trial, units often have neighbors with a mix of treatment assignments, and prior approaches to estimation use a fixed threshold to select units for use in ATE estimation:  In an extreme case, one could e.g., only consider those treated individuals for whom at least 95% of neighbors are treated, etc. This paper builds upon this line of work by proposing a data-adaptive approach for choosing these thresholds, to trade off between bias and variance.  They propose an estimator of the bias that depends on an estimate (via linear regression) of the "bias slope", the regression coefficient of the outcome on the fraction of neighbors exposed.  The main theoretical result (Theorem 4.5) characterizes the probability of choosing the MSE-minimizing threshold, and the approach is demonstrated on simulated and real data.

**Questions:**

I just have a few minor clarifying questions, to make sure that I'm understanding the paper correctly.
* Could you provide more specific detail on what is meant by cluster-level randomization here?  How are clusters determined?  Apologies if I missed this explanation in the main text.
* Is it fair to say that Theorem 4.5 assumes either unit- or cluster-level Bernoulli randomization?  If so, it might be worth calling that out as an assumption (e.g., "Assumption 4.4") a bit more explicitly.
* Does Theorem 4.5 assume Equation (1) or the slightly more restrictive form where $\psi(z, e) = g(z) + f(e)$?  That point was not entirely clear to me from the main text.

**Ethical Concerns:**

["NO or VERY MINOR ethics concerns only"]

**Final Justification:**

I am maintaining my positive score after discussion

**Limitations:**

Yes

**Quality:**

3

**Strengths And Weaknesses:**

Overall, I found the paper to be well-written and clearly explained, and I will enumerate strengths and weaknesses below.  I use (+) to indicate a strength, and (+/-) to indicate an area that is a bit more mixed.

* (+) In terms of clarity and significance, I found the problem well-motivated, and the paper clearly positioned with respect to related work in the introduction (Section 1) and Sections 2-3.  As a caveat, my expertise is in causal inference more broadly, and I am less familiar with the literature on estimation under network interference.
* (+) The algorithm and the potential outcome models are clear and well-motivated.  I appreciated the clarity of presentation here, e.g., Section 3 the note that the bias slope is only expected to be a first-order approximation, and that the method only assumes Equation (1), not the more restrictive Equation (2).
* (+) I appreciated some of the intuition provided alongside Theorem 4.5, which provides some intuition on the conditions that make it easier/harder to select the optimal MSE-reducing threshold.  It might have been nice to have results that more specifically get at the resulting MSE of the method itself (accounting for possible errors), but I'm sympathetic to the fact that such an analysis might not be very tractable given the non-smooth nature of the problem.
* (+/-) I found the technical portions of the paper to be relatively clear, modulo some points below:
    + I see an assumption (4.1) regarding positivity, but isn't there a need for assuming unconfoundedness / randomization explicitly?  For instance, that treatment assignment of $z$ does not depend on $\epsilon$?  Theorem 4.5 mentions unit-level Bernoulli randomization and cluster-level randomization, but the latter was not clearly defined in a self-contained way as far as I could tell, which led to some confusion on my part understanding the exact conditions where Theorem 4.5 holds.
    + As a minor point of clarity, I found Figure 1 difficult to read given the density of information.  I might suggest (as a minor point) to use lines for Figure 1(a) to make the trends easier to see.
* (+) I found the experiments to be a helpful addition, showcasing the fact that HT(1) and HT(0) both sometimes perform quite poorly, while the proposed approach is generally competitive with the best of the two.

---

> ### Author Rebuttal · Authors · 2025-07-27
>
> We thank Reviewer mQMx very much for the encouraging feedback and the thoughtful comments and questions. We would like to address them below.
>
> Response to Question 1: This corresponds to a cluster-level randomization design, where clusters are first defined, and then randomization is performed at the cluster level, i.e., if a cluster is assigned to treatment (resp. control), all units within that cluster receive treatment (resp. control).
>
> For Figures 2, 3, and 4: The node colors in Figure 2 reflect the clustering used in our experiments for both the graph generated from the stochastic block model (SBM) and the real-world network data. For the SBM, we used the underlying block memberships (from which the graph was generated) as the clusters. For the real dataset, we applied $\epsilon$-net clustering with $\epsilon$ = 1. In $\epsilon$-net clustering, nodes in the net are at least ε apart (in shortest path distance in the graph), and the union of their ε-radius neighborhoods (balls) covers the entire graph. These were described briefly in the captions of the figures. We apologize if this was unclear, and we are happy to improve on this clarity in the revision. We also refer the reviewer to Ugander et al. (2013) for a more detailed discussion of this clustering method.
> For the k-th power cycle graphs, clusters are formed by grouping consecutive $2k + 1$ nodes. This specific clustering choice follows the design in Ugander et al. (2013), where it is shown to achieve minimum variance. We discuss this in lines 533–534 in the Appendix.
>
> Regarding Questions 2 and 3: Thank you very much for pointing this out. This is correct, and we will make this assumption more explicit in Theorem 4.5. Corollary 4.6 does assume the slightly more restrictive form of $\psi(z,e) = g(z) + f(e)$. We will also make that clearer in the revision. Apologies for any confusion caused. Additionally, regarding clarity in Figure 1, we are happy to improve this in the revised version.
>
>  Thank you very much again for all of your insightful questions and feedback.

---

> > ### Comment · Reviewer_mQMx · 2025-08-02
> > **Thank you**
> >
> > Thank you for the response - I'll maintain my positive score

---

### Official Review · Reviewer_rfKt · 2025-06-30

**Clarity:** 3
**Significance:** 3
**Originality:** 3
**Rating:** 5
**Confidence:** 4

**Summary:**

It is well known in the causal inference literature that under network interference—where the potential outcome of a unit may depend on the treatment assignments of other units—classic estimators such as the difference in means can be biased, even under the gold standard of a randomized experiment. The typical estimand in these settings is the difference in average outcomes when everyone is treated versus when no one is treated, referred to as the average treatment effect (ATE) in the current work.

While unbiased estimation is possible using a Horvitz-Thompson estimator and a known exposure mapping, this often comes at the cost of prohibitively high variance. In this work, the authors focus on this estimator alongside a particular class of exposure mappings, where “treatment exposure” and “control exposure” are defined based on the $h$-fraction of an individual’s neighborhood assigned to treatment or control, respectively.  They propose and analyze a data-driven approach to selecting a threshold $h$ that minimizes the mean squared error of the estimator. Their method estimates bias via a simple linear regression and estimates variance using a scheme introduced by Aronow and Samii (2017). The authors analytically characterize the probability of selecting the optimal threshold under certain assumptions and empirically assess their method using simulated outcomes on both simulated and real-world networks.

**Questions:**

1.	**Possible Overloaded Notation $D$**: In Section 2.1, $D$ is introduced as a degree matrix. However, in Theorem 4.5, it appears that $D$ refers instead to a bound on the number of candidate thresholds. If that is correct, it may be worth clarifying this or using different notation to avoid potential confusion.

2.	**Possible Typo in Theorem or Proof**: In Theorem 4.5 and Corollary 4.6, the result is a bound on $\mathbb{P}(\hat{h}_n \neq h_n^{\*})$ but in Appendix A.5.1 the proof starts with $\mathbb{P}(\hat{h}_n = h_n^{\*})$.

3.	**Understanding the Bias Slope and Equation (6)**: I struggled to fully understand what the “bias slope” refers to and how it helps us interpret the bias of the Horvitz-Thompson estimator at a given threshold $h$. The sentence following Equation (6) on page 4 defines the term, but by that point, the bias slope has already been referenced several times without a clear definition. Introducing and defining it earlier would aid reader comprehension. Additionally, I had difficulty connecting the term “bias slope” to its definition. When I think about interpreting a regression coefficient, I typically understand it as: a unit change in exposure results in a $\gamma$ change in the outcome. However, the last paragraph on page 4 states that the bias slope captures “the average change in bias as $h$ varies from 0 to 1,” and I wasn’t sure how to reconcile that interpretation with the regression framing. Part of my confusion may stem from not having a strong intuitive grasp of Equation (6). While it resembles a Horvitz-Thompson-style estimator, I wasn’t clear on where it comes from or why it is a reasonable estimator for the absolute bias. A bit more elaboration or context here would be helpful.

4.	**Interpreting Corollary 4.6**: I thought the authors did an excellent job explaining the implications of Theorem 4.5, despite the complexity of the expression. However, I had more difficulty following the interpretation of Corollary 4.6, particularly the explanation in lines 244–247. For example, the sentence “Corollary 4.6 tells us… for large $n$” is helpful, but I wasn’t sure where in the expression this intuition comes from. Could the authors point out which part of the bound supports that explanation?

5.	**Figure 1(a)**: I was a little confused in the bias plot as to why there is bias when $h=1.0$. When $h=1.0$, I would expect that there is no bias but very high variance because only treated units with an entirely treated (resp. untreated) neighborhood are included under the treatment (resp. control) exposure. Is the bias there because the exposure mapping itself is misspecfied compared to the way the outcomes are generated and thus the bias comes from a misspecified exposure mapping?

6.	**Figure 4 (left)**: There is a notable drop in the magnitude of the RMSE for the last ratio value. I understood this to be referring to a regime of very strong interference. I don’t understand why suddenly all the methods have lower RMSE under this regime. Do the authors have any comments on this?

**Ethical Concerns:**

["NO or VERY MINOR ethics concerns only"]

**Final Justification:**

I had no major concerns in my original review, and the authors addressed my clarifying questions well. Other reviewers raised questions about the scope of the paper, the approach, and the assumptions, and I felt the authors did a good job of responding and highlighting the practicality of their approach. The paper is well-motivated, proposes a useful and reasonable approach to a widely used method in causal inference, and I am inclined to keep my original score.

**Limitations:**

Yes

**Quality:**

4

**Strengths And Weaknesses:**

1. **Quality**: The submission appears technically sound and is well-supported by both theoretical analysis and extensive experimental results. The methods used are appropriate and straightforward, and the work is complete. The authors do a commendable job of being both careful and honest in evaluating the strengths and weaknesses of their paper.
Specifically, they address concerns about overfitting in the appendix, with a clearly placed overview and pointer in the main text before Section 4. They are also transparent about the proposed method outperforming baselines in some settings but not all, offering a fair discussion in Section 5.1. The Discussion section further helps clarify the scope of the proposed approach, including when it is likely to be appropriate or not.

2. **Clarity**: Overall, the submission is clearly written and well-organized. The authors provide a thorough literature review and are generally careful in their explanations.
That said, a few areas would benefit from clearer or more detailed exposition—particularly around the bias estimator and the bias slope. The theoretical results, while solid, are somewhat difficult to parse due to the complexity of the expressions (involving multiple terms). I appreciated that the authors made an effort to offer intuition to help interpret these results; however, a few additional well-placed sentences could further aid reader understanding and better highlight the main takeaways. More detailed suggestions are included in the Questions section below.

3. **Significance**: The paper presents an interesting yet simple method for selecting an optimal threshold for the Horvitz-Thompson estimator of the Average Treatment Effect in a data-driven way.
Given the popularity of the Horvitz-Thompson estimator—especially when paired with cluster-randomized designs in interference settings—and the widespread use of exposure mappings based on neighborhood treatment fractions, this contribution is impactful and likely to be used or extended by others in the community.
Perhaps the biggest limitation is that such an exposure mapping is unlikely to be correctly specified in real-world applications, but this issue lies outside the scope of the current work. Regardless, since causal inference practitioners often adopt simple exposure mappings in practice, it is valuable to have a principled, data-driven approach that aims to minimize error rather than relying solely on domain knowledge to choose a threshold.

4. **Originality**: This paper offers a thoughtful combination of existing ideas in a new setting, with clear articulation of both its contributions and its relationship to prior work. They highlight important properties of the widely used Horvitz-Thompson estimator in the context of causal inference under interference. While the bias-variance trade-off is a well-known issue, the authors contribute a new data-adaptive approach to selecting a key hyperparameter—specifically, the exposure threshold—and situate this contribution within the broader literature.
The novelty here lies not in identifying the trade-off itself, but in proposing a new method for reducing error overall and situating this method in the context of the bias-variance trade-off. It is clear how this work differs from previous contributions, and to my knowledge, this particular data-driven approach has not been proposed in this specific context before. The authors are careful with their citations and transparent about how their work builds on existing techniques.

---

> ### Author Rebuttal · Authors · 2025-07-27
>
> We thank Reviewer rfKt very much for the encouraging feedback, and the thoughtful and detailed comments and questions. We would like to address them below:
>
> Response to Question 1: Thank you for pointing this out. We will be clarifying and improving the notation there.
>
> Response to Question 2: Thank you for pointing this out. We will fix the typo in the proof.
>
> Response to Question 3: Thank you for this feedback. We agree that the notion of the “bias slope” would benefit from an earlier introduction and clearer motivation, and we will incorporate this in the revision.
> To clarify the intuition: as you point out, the regression coefficient estimates the average change in the mean outcome per unit change in exposure. In our context, this also reflects the average change in bias as we move away from the boundary exposures, i.e., from 0 for control or 1 for treated, since bias arises (depending on the outcome function) when exposures deviate from these extremes. This interpretation motivates the use of the weights $e_i- 0 = e_i$ and $1− e_i$ in Equation (6), corresponding to deviations from the control and treated boundaries, respectively.
> Equation (6) defines a Horvitz-Thompson-style estimator for the bias, which integrates this regression slope over the empirical exposure distribution to yield an average (absolute) bias estimate at a given threshold.
>
> Response to Question 4: This comes from the $n(\Delta_n - \tilde{\delta})$ terms in the bounds in Corollary 4.6. Indeed, when these terms are larger, the $\exp(- (n(\Delta_n - \tilde{\delta}))$ terms are small. We hope this is helpful, and we are happy to add this explanation in the revision.
>
> Response to Question 5: Thank you for pointing this out. The issue you observed was due to this particular illustrative plot having been generated using an earlier version of the code, with significantly fewer Monte Carlo trials. This was an oversight; all other plots and results in the paper had already been updated to use 1,000 Monte Carlo trials for bias computation and 10,000 for exposure probability estimation. We have now regenerated this specific plot using the updated setup. As expected, the bias at threshold 1 is numerically close to zero across all $\gamma/\beta$ ratios, consistent with your intuition, and bias remains near zero for $\gamma/\beta = 0$. We will replace the outdated figure in the revision to ensure consistency with the rest of the results.
>
> Response to Question 6:
> Thank you for pointing this out. Upon reviewing the code used to generate this plot, we discovered an inconsistency: the ATE used to normalize the RMSE was inadvertently computed using a $\gamma/\beta$ ratio of $10^6$, while the RMSE values themselves were computed using the correct ratio of $10^4$. This mismatch led to an artificially low normalized RMSE for the final data point.
> The value $\gamma/\beta = 10^6$ was used elsewhere in the cluster-randomized setting to illustrate the effects of stronger interference, as cluster randomization helps control interference bias. Unfortunately, that parameter setting was mistakenly carried over when computing the ATE for this plot.
> We have since corrected the figure, and the updated version behaves as expected: the normalized RMSE no longer drops sharply for the final ratio and reflects a consistent trend across all $\gamma/\beta$ ratios. We apologize for this oversight and will include the corrected figure in the revision. We have verified that this issue is isolated to this plot and does not affect any other results in the paper.
>
> Once again, we very much appreciate the reviewer's thoughtful and detailed feedback.

---

> > ### Comment · Reviewer_rfKt · 2025-08-04
> > **Thank you!**
> >
> > Thank you for the thoughtful responses! I have no further questions or concerns.

---

### Official Review · Reviewer_Szy4 · 2025-07-03

**Clarity:** 3
**Significance:** 2
**Originality:** 2
**Rating:** 3
**Confidence:** 4

**Summary:**

This work considers estimating the average treatment effect (ATE) in the presence of network interference. A common approach to obtain unbiasedness is to define an exposure mapping that identifies units not subject to interference, and then apply a Horvitz–Thompson (HT) estimator restricted to those units. In such a case, the HT estimator would compare outcomes of treated and control units whose neighborhoods are sufficiently homogeneous (according to threshold $h$), weighting by their assignment probabilities. The core difficulty is that the threshold $h$ is typically fixed a priori and hard to choose while a mis-specified $h$ can bring bias or inflate variance. The paper’s motivation is to adaptively choose the exposure threshold rather than rely on a fixed one to best balance interference bias and statistical variance.

**Questions:**

- Why is the Horvitz–Thompson estimator preferred over other estimators (e.g., Hajek, AIPW, regression-adjusted)?
- How does your method compare to direct modeling approaches that predict outcomes as a function of both treatment and exposure?

**Ethical Concerns:**

["NO or VERY MINOR ethics concerns only"]

**Limitations:**

Only considers Horvitz–Thompson (HT) estimator for causal inference under network interference.

**Quality:**

3

**Strengths And Weaknesses:**

Strengths
- adaptively choosing the exposure threshold for HT estimator is well motivated.
- The paper’s goal is to select $h$ that minimizes the MSE, which makes sense and leads to the bias variance trade-off.
- Bias and variance are estimated in a data-driven way to select the optimal threshold.


Weakness
- The method estimates bias based on a linear regression of outcomes on exposure
- Estimating inclusion probabilities can be challenging when the network is small or has high degrees
- AdaThresh needs a range of exposure thresholds to be well-populated with units. Otherwise, many thresholds may include too few units to reliably estimate HT statistics or regression slopes.

---

> ### Author Rebuttal · Authors · 2025-07-27
>
> We thank Reviewer Szy4 for the thoughtful review and questions. We would like to address the questions and concerns the reviewer has posed below.
>
> Response to Question 1: The Horvitz-Thompson estimator is used heavily in practice! Motivated by this, we sought to address our framework through the setting of the Horvitz-Thompson estimator. Our framework, of course, could be adapted for the Hajek, Difference-in-Means, AIPW etc. estimators. We formulate the adaptively-thresholded estimator for the Difference-in-Means estimator, which is a special case of the Hajek estimator, in the Appendix (Section A.7.5.). A similar adaptation to the Hajek estimator could be used. We note however, that the Hajek estimator is only approximately unbiased at the true threshold. Therefore, characterizing the exact bias-variance tradeoff formally would be more complex compared to the Horvitz-Thompson estimator. We discuss this in lines 74-80 of the paper.
>
> Response to Question 2:
> We appreciate the curiosity of how the performance of our approach compares to direct modelling approaches. This, however, is not the focus of our paper. Instead, our work focuses on an adaptive optimal threshold for a specific estimator class decided upon by the practitioner a priori. In particular, our goal is to show that there is something to be gained when using an adaptive estimator, instead of a fixed estimator like HT(0) or HT(1), as in our approach. The only other viable adaptive approach that exists for that purpose is Lepski’s method which we had adapted to our settings. Therefore, we believe that the useful benchmarks for comparisons are the estimators adapted from Lepski’s method, and the fixed Horvitz-Thompson estimators.
>
> We would also like to emphasize that, while we use linear regression to obtain a first-order approximation of how interference bias varies with exposures, which is similar in spirit to Baird et al. (2018), we go further by pairing this with a Horvitz-Thompson-style bias estimator. This approach uses the estimated regression coefficients in conjunction with exposures, enabling the bias estimator to account for the interaction between regression structure and exposure distribution present in the data.
>
> We hope that our response has provided more clarity on the motivations and the focus of our work. We thank the reviewer once again for the thoughtful feedback, and we hope that this discussion has enhanced your perspective on our paper.

---

### Decision · Program_Chairs · 2025-09-17

**Decision:**

Accept (poster)

**Comment:**

The paper presents a new method for estimating the bias and variance of the Horvitz-Thompson estimator, under interference, with the assumption that at least h-fraction of a unit's neighbors share its treatment. The threshold h itself is selected by their method from several candidates. The method works by "leveraging a first-order approximation, specifically, linear regression of potential outcomes on exposures".

For the most part, the reviewers praised the paper, considering the method well motivated and novel. The writing is clear and insightful, with a few questions on the figures and result interpretation being clarified during the discussion period.


The main criticism from one of the reviewers is that the work is limited to the HT estimator. However, given how widely used the estimator is, and that many other papers in the area focus on modifications and analysis for a single estimator, I evaluate this paper to be well scoped.

All in all, the paper provides an elegant solution to a known problem, and is a valuable contribution to the field.